# Therapeutic vulnerability to PARP1,2 inhibition in *RB1*-mutant osteosarcoma

Georgia Zoumpoulidou [1], Carlos Alvarez-Mendoza [1], Caterina Mancusi[1], Ritika-Mahmuda Ahmed [1], Milly Denman[1], Christopher D. Steele[1], Maxime Tarabichi[2,3], Errin Roy [1], Lauren R. Davies [1], Jiten Manji[1], Camilla Cristalli [4], Katia Scotlandi [4], Nischalan Pillay [1,5], Sandra J. Strauss[1,6] & Sibylle Mittnacht [1✉]

Loss-of-function mutations in the *RB1* tumour suppressor are key drivers in cancer, including osteosarcoma. *RB1* loss-of-function compromises genome-maintenance and hence could yield vulnerability to therapeutics targeting such processes. Here we demonstrate selective hypersensitivity to clinically-approved inhibitors of Poly-ADP-Polymerase1,2 inhibitors (PARPi) in RB1-defective cancer cells, including an extended panel of osteosarcoma-derived lines. PARPi treatment results in extensive cell death in RB1-defective backgrounds and prolongs survival of mice carrying human RB1-defective osteosarcoma grafts. PARPi sensitivity is not associated with canonical homologous recombination defect (HRd) signatures that predict PARPi sensitivity in cancers with *BRCA1,2* loss, but is accompanied by rapid activation of DNA replication checkpoint signalling, and active DNA replication is a prerequisite for sensitivity. Importantly, sensitivity in backgrounds with natural or engineered RB1 loss surpasses that seen in *BRCA*-mutated backgrounds where PARPi have established clinical benefit. Our work provides evidence that PARPi sensitivity extends beyond cancers identifiable by HRd and advocates PARP1,2 inhibition as a personalised strategy for *RB1*-mutated osteosarcoma and other cancers.

[1] UCL Cancer Institute, University College London, London, UK. [2] The Francis Crick Institute, London, UK. [3] Institute for Interdisciplinary Research, Université Libre de Bruxelles, Brussels, Belgium. [4] Experimental Oncology Laboratory, IRCCS Istituto Ortopedico Rizzoli, Bologna, Italy. [5] Department of Histopathology, Royal National Orthopaedic Hospital, Stanmore, London, UK. [6] London Sarcoma Service, University College London Hospitals Foundation Trust, London, UK. ✉email: s.mittnacht@ucl.ac.uk

Biallelic mutations targeting *RB1* are prominently associated with difficult to treat cancers, including osteosarcoma. Osteosarcoma is the most common primary human bone malignancy. More than half of cases arise in children and young adults, with disproportionate contribution to cancer death in these age groups[1]. Aggressive multimodal treatment involving combination chemotherapy substantially increases survival.

However, less than 30% of patients diagnosed with metastatic disease show long-term response; and relapse and treatment associated toxicity in patients diagnosed with localised disease remain chief concerns[1–4].

Emerging osteosarcoma genomics data reveal the prominent presence of deleterious mutations in the known tumour suppressors *TP53*, *RB1*, *ATRX* and *CDKN2A*[5–7]. *RB1* mutations are

**Fig. 1 Differential PARPi sensitivities in RB1-defective and RB1-normal osteosarcoma-derived tumour cell lines.** Cells seeded in 96-well plates were treated with PARPi at concentrations as indicated. Cell viability was determined 5 days following inhibitor addition using resazurin-reduction. Concentration-response curves for **a** olaparib, **c** niraparib, or **e** talazoparib. Curves shown are representative for $n = 3$ (**a**) or $n = 2$ (**c**, **e**) biologically independent datasets. Data points represent the mean ± SD of parallel triplicate values, plotted relative to vehicle-treated controls, set to 100%. Osteosarcoma-derived RB1-defective (red), RB1-normal (black) and pancreatic ductal carcinoma *BRCA2*-mutant CAPAN1 (blue). Scatter plots summarising area-under-the-curve (AUC) values deduced from dose response curves for RB1-defective (red) or RB1-normal (black) osteosarcoma lines or *BRCA2*-mutant CAPAN1 (blue), treated with **b** olaparib, **d** niraparib, or **f** talazoparib, for $n = 2$ biologically independent datasets, respectively. Bars depict median values ±95% confidence interval (CI), *$p < 0.05$, **$p < 0.01$, ****$p < 0.0001$, using a two-tailed Mann–Whitney test, $p$ (**b**) $< 0.0001$, $p$ (**d**) $= 0.0065$ and $p$ (**f**) $= 0.0127$. **g**, **h** Pearson product moment correlation measuring the strength of a linear association between AUC data for olaparib and talazoparib (red), olaparib and niraparib (blue) or olaparib and a second olaparib dataset (black), **g** RB1-defective osteosarcoma lines and **h** RB1-normal osteosarcoma lines. Tables showing Pearson's correlation coefficient and $p$ values for two-tailed tests. **i** Concentration-response curve for PARPi Veliparib depicting mean of three parallel samples relative to vehicle-treated controls. Data reflect the mean ± SD of parallel triplicates for one of $n = 2$ biologically independent datasets. **j** Scatter plot summarising AUC values for $n = 2$ biologically independent datasets, ($p < 0.0001$). Bars depict median and ±95% confidence interval (CI), ****$p < 0.0001$, using a two-tailed Mann–Whitney test. **k** Symbols and names for cell lines used. Osteosarcoma-derived RB1-defective (red), RB1-normal (black), pancreatic ductal carcinoma BRCA2-mutant CAPAN1 (blue). **l** Immunoblotting analysis assessing the expression of RB1 in osteosarcoma-derived cell lines. GAPDH was used as loading control. Source data are provided as a Source Data file.

seen in 40–60% of sporadic osteosarcoma[5–7], making it the second most commonly mutated gene in this disease after *TP53*. Studies of osteosarcoma genomic evolution invariably report *RB1* mutations as early, truncal events[8,9] and germline mutations in *RB1* increase the risk of osteosarcoma development[10], denoting a causal role of *RB1* defects during disease initiation. Notably, various sources, including a recent systematic review, report association of *RB1* mutation with poor prognosis including high risk of metastasis[11], paralleling observations in other cancers with *RB1* involvement[12,13] and indicating a clear unmet clinical need in patients with *RB1*-mutant osteosarcoma.

While conventional combination chemotherapy has remained standard of care for osteosarcoma irrespective of presentation or genotype[3], targeted agents including multitargeted tyrosine kinase inhibitors show efficacy in early phase clinical trials and may offer additional options in relapsed disease, albeit with cost of significant treatment-related toxicity[14]. Personalised, biomarker-informed opportunities have been identified by preclinical work for various gain-of-function events[5,7,15], indicating targeted, genome-informed treatment could provide future solutions in osteosarcoma. However, opportunities identified by the highly prevalent loss-of-function events, including *TP53* and *RB1*, have not been reported.

The *RB1*-encoded protein (RB1) is a negative regulator of the cell cycle but has been ascribed other functions[16]. RB1 defects in cells cause complex changes including anomalies in DNA double-strand-break (DDSB) repair[17–19] and mitotic fidelity[20]. Such DNA metabolic alterations raise the possibility that synthetic lethal opportunities may exist involving therapeutics known to interact with defective DNA repair or mitosis.

Based on assessment of an extended osteosarcoma-focused cell line panel, we here report selective sensitivity of RB1-defective osteosarcoma to inhibitors of poly-(ADP-ribose)-polymerase1,2 (PARPi). PARP1,2 enzymes have complex roles in DNA single-strand-break (DSSB) repair, transcription and replication[21]. PARP1,2 inhibition is selectively lethal in cancer with mutation in the *BRCA1,2* tumour suppressors, causing HRd[22], and multiple PARPi have regulator approval for the treatment of HRd and/or *BRCA1,2*-mutated ovarian, breast and pancreatic cancers, with recent FDA breakthrough status in castration-resistant prostate cancer[23,24].

We here document highly penetrant PARPi hypersensitivity following from *RB1* mutation, with dose sensitivity comparable to that caused by *BRCA1,2* mutation. We validate the involvement of RB1 defects in this response and document single-agent PARPi efficacy in a preclinical model of *RB1*-mutant osteosarcoma. Our work proposes a genome-led strategy for treatment of osteosarcoma, involving stratified use of PARP1,2 targeting therapeutics.

## Results

**PARPi sensitivity in RB1-defective osteosarcoma tumour cell lines.** To identify therapeutically exploitable vulnerability linked to deleterious *RB1* mutation we assessed the sensitivity of histotype-matched cancer cell line pairs differing in *RB1* mutation status to clinical candidate agents that target DNA metabolic processes.

Day 5 viability assessments involving resazurin-reduction revealed consistent hypersensitivity to the PARPi olaparib in RB1-defective compared to RB1-normal lines (Supplementary Fig. 1a–c). A strong association between olaparib sensitivity and RB1 status extended to a poly-cancer cell line panel, with median area-under-the-curve (AUC) values significantly lower in RB1-defective compared to RB1-normal lines (Supplementary Fig. 1d–f), indicative that RB1 status in cancers is associated with, and may predict, hypersensitivity to PARPi.

Significantly, the increased dose sensitivity to olaparib extended to a broad osteosarcoma-focussed cell panel (Fig. 1a), yielding a highly significant differential median sensitivity assessed using AUC values (Fig. 1b) in lines with known *RB1*-mutant status and/or lacking detectable RB1 expression (Fig. 1l), compared to RB1-normal lines. Significantly, median sensitivity in the RB1-defective group was comparable to that of the pancreatic cancer line CAPAN1, known for profound PARPi sensitivity due to defective *BRCA2*[25] that we included to benchmark clinically relevant response levels.

Similar results were obtained using the clinically approved but structurally unrelated PARPi niraparib (Fig. 1c, d) and talazoparib (Fig. 1e, f). Both yielded significantly increased median sensitivity for the RB1-defective compared to the RB1-normal osteosarcoma group, with sensitivities across the RB1-defective group greater than, or closely matching that of *BRCA2*-mutated CAPAN1 (Fig. 1b, d, f). High correlation coefficients and highly significant linear correlations were obtained comparing repeat assessments of the same PARPi (Pearson $r = 0.92$, $p < 0.0001$ for olaparib, Pearson $r = 0.98$, $p < 0.001$ for niraparib and talazoparib), (Supplementary Fig. 1g–i), indicating reliability of the analysis. Importantly, highly significant linear correlations were obtained comparing different PARPi, (Fig. 1g, h), indicating that their shared activity of targeting PARP1,2 underlies the sensitivity profiles observed.

A significant association between sensitivity and *RB1*-defect was also observed using veliparib, a PARPi that inhibits PARP1,2 catalysis but lacks the ability to trap PARP1,2 enzymes on

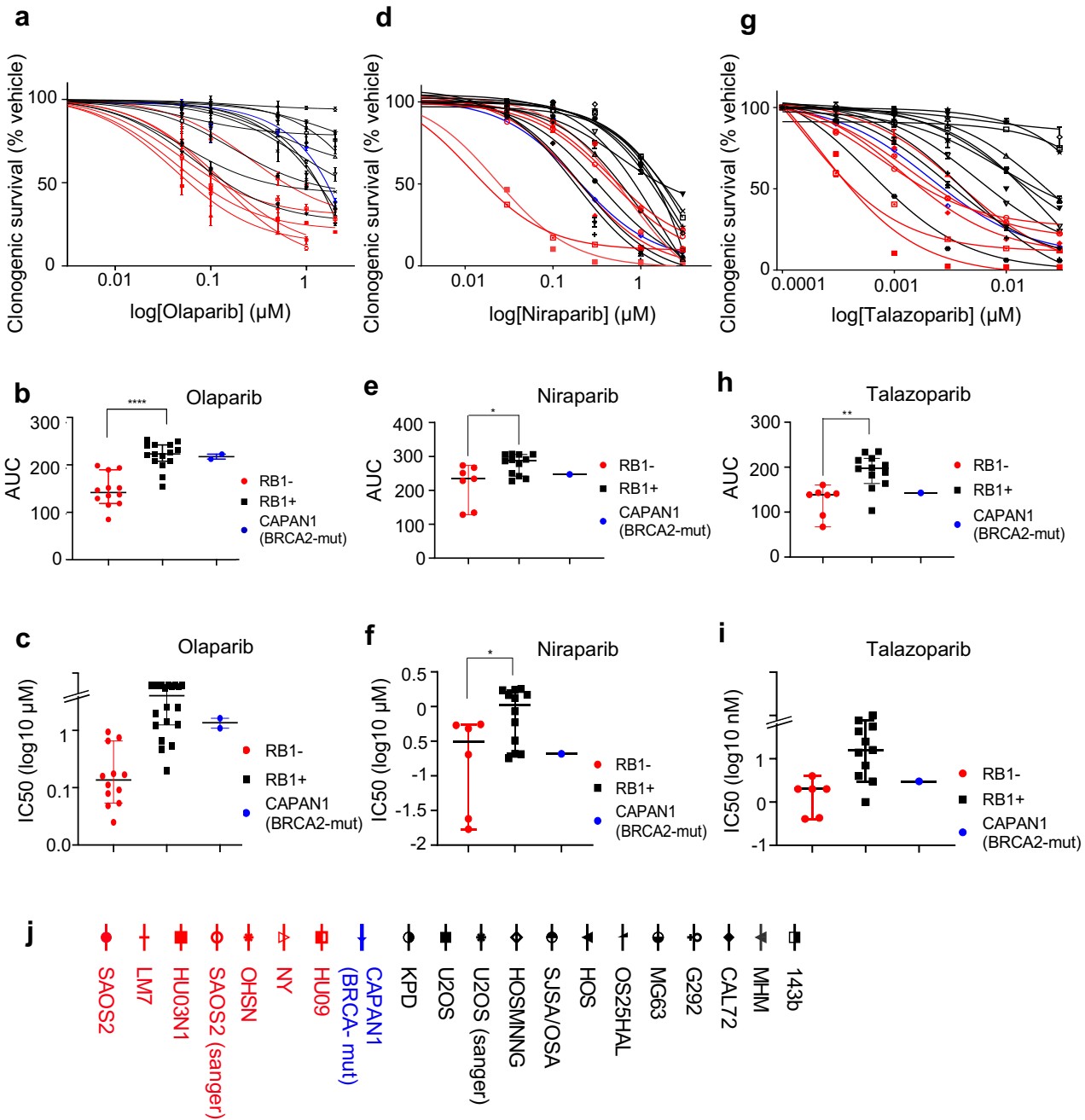

**Fig. 2 Effect of PARP inhibition on clonogenic survival.** Cells were seeded at low density into six-well plates followed by treatment with increasing concentrations of PARP inhibitors or vehicle. Colonies arising were stained using crystal violet dye. Clonogenic survival was quantified using dye extraction. Clonogenic survival concentration-response curves for RB1-defective (red) or RB1-normal (black) osteosarcoma and *BRCA2*-mutant CAPAN1 (blue) after treatment with **a** olaparib, **d** niraparib, or **g** talazoparib. Data reflect the mean ± SD of parallel duplicates from one dataset. Scatter plots comparing AUC values for RB1-defective or RB1-normal osteosarcoma lines and *BRCA2*-mutant CAPAN1 treated with **b** olaparib, **e** niraparib or **h** talazoparib summarising data for $n = 2$ (**b**) or $n = 1$ (**e**, **h**) biologically independent dataset. Bars depict median ±95% CI. *$p < 0.05$, **$p < 0.01$, ****$p < 0.0001$ calculated using two-tailed Mann–Whitney tests, $p$ (**b**) $< 0.0001$, $p$ (**e**) $= 0.022$ and $p$ (**h**) $= 0.0012$. **c**, **f**, **i** Scatter plots depicting IC50 values deduced from dose response data in **b**, **e** and **h**. Bars depict median ±95% CI; $p$ (**f**) $= 0.032$*, calculated using two-tailed Mann–Whitney tests. **j** Symbols and names for cell lines used. Source data are provided as a Source Data file.

damaged chromatin[26,27], (Fig. 1i, j), with significant correlation comparing repeat assessments (Supplementary Fig. 1j). However, the differential in sensitivity was small and the inhibitor concentration required to affect viability high. While consistent with an increased dependency on PARP1,2 catalysis, these results indicate that PARP trapping may be an important mechanistic determinant for single-agent potency in *RB1*-mutant osteosarcoma, as is known for *BRCA1,2*-mutated cancers[22].

Clonogenic assays, scoring for the ability of cells to form colonies, confirmed selective PARPi hypersensitivity in RB1-defective osteosarcoma for all three PARPi (Fig. 2a–j, raw data Supplementary Fig. 2a–c) with half maximal inhibitory concentrations (IC50) for RB1-defective osteosarcoma matching or below that determined for *BRCA2*-mutant CAPAN1, and differentials in median IC50 value comparing RB1-normal and RB1-defective groups of 14-fold (olaparib), fivefold (niraparib)

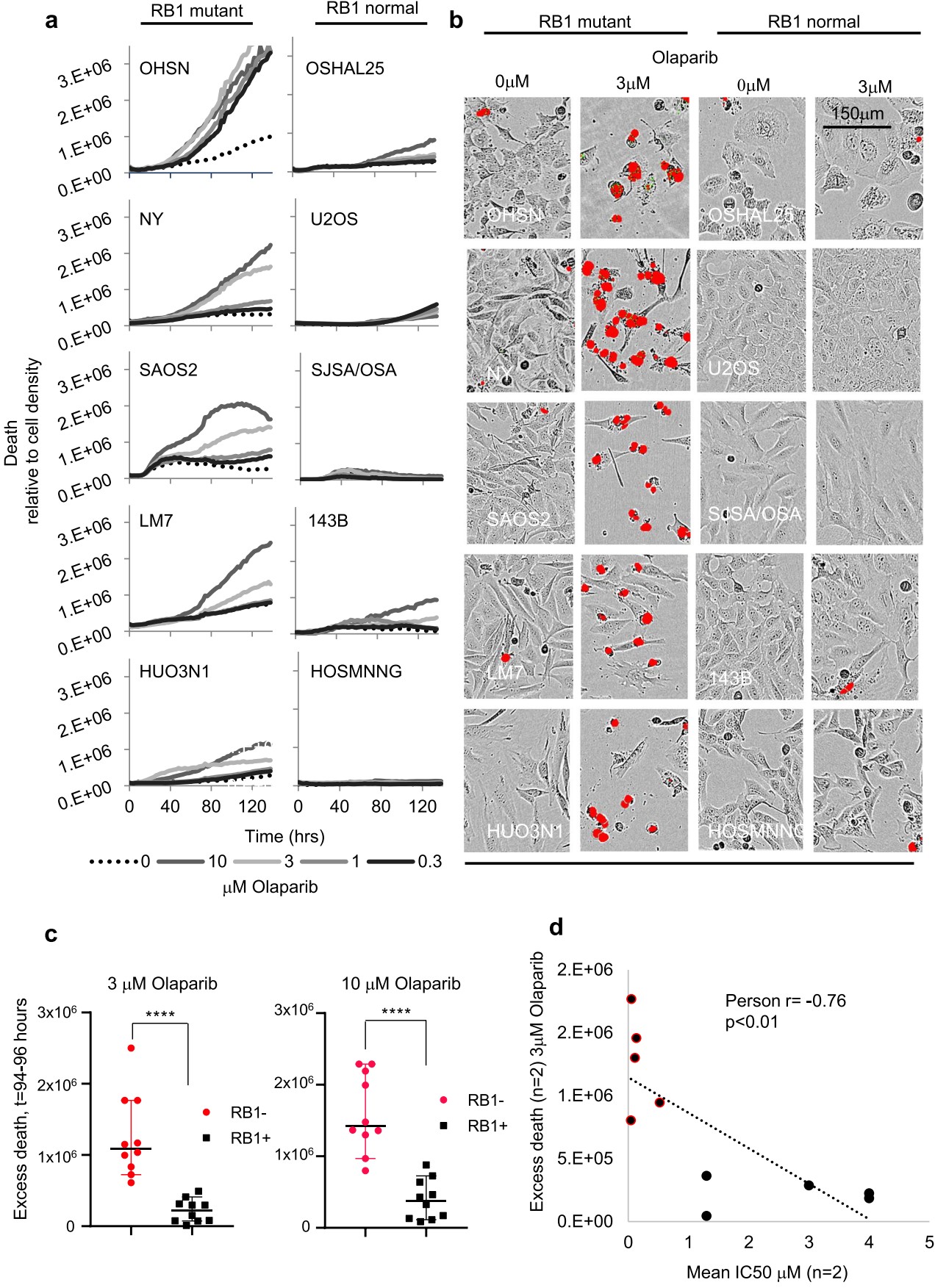

**Fig. 3 Cellular effects of PARPi treatment.** RB1-defective and RB1-normal osteosarcoma lines seeded in 96-well plates were treated with PARPi olaparib at concentrations indicated, then subjected to time-lapse microscopy in the presence of SYTOX[TM] death-dye. Images were taken every 2 h, recording phase contrast and death-dye fluorescence. **a** Death-dye incorporation over time relative to cell density in RB1-defective (left) and RB1-normal (right) osteosarcoma cancer lines. **b** Raw images 96 h post inhibitor addition, depicting phase contrast superimposed with death-dye fluorescence. **c** Mean death above vehicle (excess death) 94–98 h after olaparib addition. Olaparib concentration was as indicated. **d** Pearson product moment correlation assessing the strength of a linear association between mean excess death at 94–98 h and IC50 (determined using clonogenic survival assays) for the respective cell lines. Pearson's correlation coefficient (Pearson r) and p values for two-tailed tests are indicated. Data (**a**, **b**) are exemplary for two or more biologically independent datasets or (**c**, **d**) summarise data for $n = 2$ biologically independent datasets ($p < 0.0001$). Bars depict median ±95% CI. ****$p < 0.0001$, two-tailed Mann–Whitney test. Source data are provided as a Source Data file.

and eightfold (talazoparib) (Fig. 2c, f, i and Supplementary Table 1). Superior selectivity of olaparib over niraparib was previously observed in HRd cancers and may relate to differences in off-target activity of these different agents[28].

Collectively the data provide evidence that RB1 status is a predictor of single-agent PARPi sensitivity in osteosarcoma, with sensitivity levels comparable to that of *BRCA2*-mutant cells.

**PARPi-induced cell death in RB1-defective osteosarcoma.** To understand how PARPi act to reduce cell viability in RB1-defective osteosarcoma we performed time-lapse microscopy using medium containing SYTOX[TM] death-dye, which marks cell death. Treatment with olaparib yielded a concentration-dependent increase in death-dye incorporation compared to vehicle-treated controls (Fig. 3a, b) accompanied by widespread cytopathic effects (Fig. 3b) in RB1-defective but not RB1-normal osteosarcoma lines.

Increased death-dye incorporation and cytopathic effects became evident 40 and 60 h after PARPi addition. Statistics comparing death above vehicle (excess death) at 94–96 h identified a highly significant differential between the RB1-mutant and RB1-normal group with a strong and significant inverse correlation between death and the IC50 for the respective lines, consistent with a link between olaparib-induced death and antiproliferative response (Fig. 3c, d).

Corroborative results were obtained using talazoparib or niraparib, with concentration-dependent death in RB1-mutant but not RB1-normal osteosarcoma, and similar time to onset (40–60 h) regardless of PARPi used (Supplementary Fig. 3a–c). Together these results are consistent with an enhanced sensitivity to PARPi in *RB1*-mutant osteosarcomas and identify rapid cell death as a key consequence of PARPi exposure in osteosarcoma cells with this genetic defect.

**PARPi sensitivity is a consequence of RB1 deficiency.** To investigate if selective PARPi sensitivity in *RB1*-mutant osteosarcomas is a consequence of RB1 loss, we depleted RB1 in RB1-normal osteosarcoma lines CAL72 or 143b using multiple *RB1*-targeting small-hairpin RNAs (shRNA) (Fig. 4 and Supplementary Fig. 4).

Clonogenic survival assessments revealed a significant increase in olaparib sensitivity of the various RB1-depleted lines compared to unmodified CAL72 or empty vector controls (Fig. 4b, c and Supplementary Fig. 4a), yielding IC50 values for the RB1-depleted group in the submicromolar range and a differential in median IC50 compared to controls of >10-fold (Fig. 4d and Supplementary Table 2). A significantly greater sensitivity of RB1-depleted CAL72 was also observed using niraparib (Fig. 4e–g, Supplementary Table 2, and Supplementary Fig. 4b), although with a smaller differential in median IC50 between groups (6–7-fold), consistent with observations comparing naturally RB1-defective and RB1-normal osteosarcoma lines.

Congruent outcomes were obtained in day 5 viability assays, documenting significantly increased olaparib sensitivity in the

RB1-depleted CAL72 (Supplementary Fig. 4c, d) or 143b (Supplementary Fig. 4e–g) compared to their respective maternal lines or derivatives modified using empty vector or vector encoding an irrelevant shRNA targeting the human haemoglobin A (HBA).

Consistent with the reduced day 5 viability, analysis of RB1-depleted CAL72 using time-lapse microscopy revealed a significant rise in cell death compared to controls, that progressively increased with increasing olaparib (Fig. 4h, j) or talazoparib (Fig. 4i, k) concentrations. Collectively, these data provide strong evidence that RB1 loss is causative of the increased hypersensitivity of *RB1*-mutant osteosarcomas to PARP1,2 inhibition.

**Determinants of PARPi sensitivity in RB1-defective osteosarcoma.** Since PARPi hypersensitivity in cancers is caused by BRCAness/ HRd[29], we determined if RB1 loss may yield BRCAness/ HRd. The inability of cells to recruit the DNA recombinase RAD51 to DDSBs is regarded an indicator of BRCAness/ HRd[30,31]. We therefore quantified RAD51 recruitment in RB1-defective osteosarcoma lines using ionising radiation (IR) to induce DDSBs (Fig. 5a–c). To benchmark response, we included HR and RAD51 recruitment defective CAPAN1, and HR and RAD51 recruitment competent colorectal carcinoma HT29 cells.

We observed a significant DNA damage-dependent RAD51 recruitment, evidenced by increased numbers of cells with >15 RAD51 foci, and a significant increase in foci numbers per cell in all RB1-defective osteosarcoma lines except one, LM7. LM7 have previously been reported as RAD51 recruitment defective[15], thought to be linked to reduced expression of multiple HR components. As expected, inability of DNA damage-dependent RAD51 recruitment was seen in CAPAN1, while substantive gain in RAD51 positive cells and significant increase in foci numbers was seen in HR competent HT29 (Fig. 5a–c).

We further assessed the impact of the 53BP1 loss (Supplementary Fig. 5b–n). 53BP1 ablation broadly abolishes PARPi sensitivity consequent to BRCAness/ HRd[32–35]. While shRNA-mediated 53BP1 depletion led to overt resistance to olaparib in *BRCA1*-mutated breast cancer SUM149 cells, no effect was seen in RB1-defective osteosarcoma OHSN or SAOS2 cells. These results are consistent with a view that molecular events mechanistically distinct from BRCAness/ HRd are responsible for the hypersensitivity in osteosarcoma with RB1 loss.

Finally, we performed genomic scar analysis (Fig. 5d) making use of exome sequence and SNP data publicly available for ten of the osteosarcoma lines and whole genome sequence that we newly generated for the remaining 7. We used these data to derive scarHRD scores[36], a computation strategy based on quantification of telomere allelic imbalance, loss of heterozygosity and large-scale transitions, that together provide a DNA-based measure of BRCAness/HRd[36,37] able to identify BRCAness in patients without *BRCA1,2* mutation in clinical contexts[38]. This analysis yielded scarHRD scores clearly lower than those detected for *BRCA1,2*-mutated cell lines for all osteosarcoma lines and

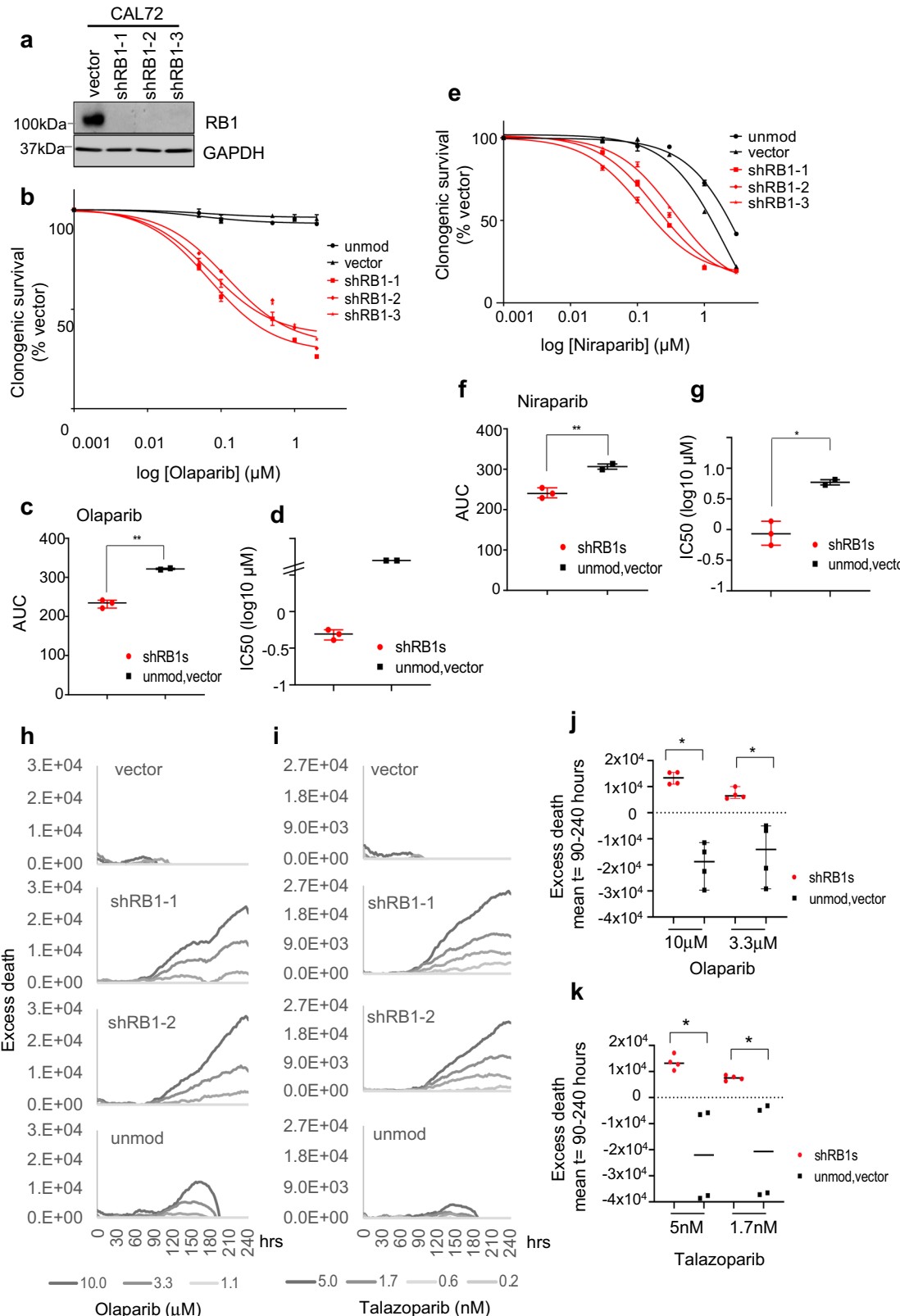

absence of a significant difference between RB1-defective and RB1-normal osteosarcoma groups (Fig. 5d, e).

HRd in cancers is further associated with a somatic single nucleotide mutational signature identified as single-base substitution signature 3 (SBS03)[39]. While this signature was not robustly detected in the cell line sequences even if they contained *BRCA1,2* mutation and confirmed HRd (Fig. 5d), most probably because the signature is flat and cannot be reliably identified from exome data, analysis of published osteosarcoma whole genome data[7] confirmed *RB1* defects are not significantly associated with HRd exposure using this parameter (Supplementary Fig. 5a). Namely, SBS03, and with this evidence for HRd exposure was not detectable in 5 of 10 tumours with *RB1* mutation and HRd exposure had no significant linkage to *RB1* mutational status.

**Fig. 4 PARPi response in RB1-normal osteosarcoma following RB1 depletion.** RB1-normal osteosarcoma CAL72 cells were infected with lentivirus vector encoding different *RB1*-targeting shRNAs (shRB1-1, shRB1-2 or shRB1-3) or empty vector backbone (vector) or were left unmodified (unmod.).
**a** Immunoblot analysis documenting RB1 expression. GAPDH was used as a loading control. Clonogenic survival analysis depicting concentrations-effect curves (**b**, **e**), scatter plots for AUC (**c**, **f**) and IC50 values (**d**, **g**) for cells treated with olaparib (**b**, **c**, **d**) or niraparib (**e**, **f**, **g**). CAL72 with modifications as indicated were seeded into six-well plates and treated with PARPi at concentrations as indicated. Data reflect the mean ± SD of parallel duplicate wells for one of $n = 2$ biologically independent datasets. Bars depict median ±95% CI. *$p < 0.05$, **$p < 0.01$, calculated using two-tailed Mann–Whitney tests, $p$ (**c**) = 0.0015, $p$ (**f**) = 0.0083 and $p$ (**g**) = 0.0026. **h**–**k** Time-lapse microscopy assisted fate assessment. CAL72 modified as indicated were treated with PARPi and monitored for death-dye incorporation over time. **h**, **i** Excess death above vehicle over time. **j**, **k** Mean excess death between 90 and 240 h. Graphs represent one (**h**, **i**) of $n = 2$ biologically independent datasets or **j**, **k** summarise results for $n = 2$ biologically independent datasets Bars depict median ±95% CI. *$p < 0.05$, calculated using two-tailed Mann–Whitney tests. $p$ (**j**) = 0.028 for 10 μM, $p$ (**j**) = 0.0286 for 3 μM, $p$ (**k**) = 0.028 for 10 μM, $p$ (**k**) = 0.029 for 3 μM. Source data are provided as a Source Data file.

Together these data argue that defects in *RB1* are not associated with canonical features of HRd/ BRCAness and hence that PARPi sensitivity in RB1-defective osteosarcoma is mechanistically distinct from and not explained by outright inability to engage HR-based DNA repair.

**Platinum sensitivity in RB1-mutated osteosarcoma.** PARPi sensitivity in *BRCA1,2*-mutated cancer is paralleled by hypersensitivity to platinum drugs, and platinum drug sensitivity is a predictor of BRCAness/ HRd. Importantly, platinum drugs are an important component of clinical care in osteosarcoma. We therefore assessed if RB1 status, that our work shows predicts PARPi sensitivity, might likewise predict platinum sensitivity.

However, assessment of response to cisplatin across the various osteosarcoma lines using clonogenic survival (Fig. 6a–c, Supplementary Fig. 6a, Supplementary Table 1) or day 5 viability (Supplementary Fig. 6b, c) revealed no significant difference in AUC or IC50 value distributions between groups. Notably, median sensitivities closely matched that for *BRCA2*-mutated, cisplatin-hypersensitive CAPAN1[40], indicating high platinum sensitivity across osteosarcoma lines, irrespective of status and PARPi sensitivity.

To assess if RB1 defects could cause platinum sensitivity we made use of the RB1-depleted CAL72. While unmodified CAL72 had modest cisplatin sensitivity (Supplementary Table 2, IC50 > 1 μM), a significant and substantive sensitivity increase was seen in RB1-depleted CAL72, using clonogenic activity (Fig. 6d–f, Supplementary Table 2, Supplementary Fig. 6d) or day 5 viability (Supplementary Fig. 6e, f). Hence, although platinum sensitivity is widespread amongst the established osteosarcoma lines and here is not predicted by RB1 status, these latter data argue that RB1 defects, alike *BRCA1,2* defects, increase platinum sensitivity.

**PARPi activate DNA replication checkpoint response in RB1-defective osteosarcoma.** To begin to understand what causes the PARPi hypersensitivity in *RB1*-mutant cancer cells we assessed DDSB-damage response activation in RB1-defective and RB1-normal osteosarcoma cell lines. PARP inhibition prevents the ligation of DSSBs and traps PARP complex on these lesions, leading to DDSB once cells move into S-phase, which cannot be appropriately resolved in PARPi sensitive cells with HRd[41], reviewed in refs. [42,43].

To assess if DDSBs arise and may selectively accumulate in RB1-mutant cells, we measured the level of the DDSB repair histone marker γH2AX using immunohistochemistry (Fig. 7a, b and Supplementary Fig. 7a–c).

We observed a robust and significant rise in γH2AX-positive cells following treatment with PARPi olaparib in two different RB1-defective osteosarcoma lines (Fig. 7a), seen within 2 h but increasing with time of treatment. Signals in cells positive for γH2AX were confined to the cell nucleus with characteristic

speckled appearance, comparable in distribution and intensity to those caused by IR (Supplementary Fig. 7a). No significant increase in γH2AX-positive cells compared to vehicle treatment was observed in two RB1-normal osteosarcoma lines albeit γH2AX-positive cells significantly increase following IR exposure (Fig. 7a and Supplementary Fig. 7b). Importantly, PARPi treatment significantly increased γH2AX positivity following RB1 depletion in RB1-normal osteosarcoma CAL72 (Fig. 7b) or RB1-normal osteosarcoma 143B cells (Supplementary Fig. 7d–g), yielding a significant increase in the percentage of γH2AX-positive cell following depletion compared to their respective unmodified maternal lines (Supplementary Fig. 7c, h). No significant increase was seen in cells that expressed irrelevant control shRNA targeting HBA. These results indicate that canonical DDSB-damage signalling ensues in response to PARP inhibition of RB1-defective osteosarcoma, with direct evidence that RB1 loss is a prerequisite and causative in this response.

γH2AX may signify activation of distinct DNA damage response pathways, notably, ATM, activated in response to DDSB, or ATR, activated in response to DNA replication impairment. To delineate which of these pathways may be activated we scored for the activating phosphorylation of checkpoint kinase CHK1, targeted by ATR, and CHK2, selectively linked to ATM signalling[44]. Quantitative immunoblot analysis revealed a prominent and significant increase in CHK1 phosphorylation following PARPi treatment of the RB1-defective OHSN (Fig. 7c, d), equal to or surpassing that observed in response to IR in the same cells (Fig. 7d). Using the same lysates, only a modest increase in phosphorylation of CHK2 was observed, despite strong phosphorylation of CHK2 following IR (Fig. 7c, e). PARPi treatment failed to significantly increase phosphorylation of CHK1 in the RB1-normal CAL72 (Fig. 7f–h). However, a significant increase in CHK1 activation arose in RB1-ablated CAL72 (Fig. 7l–n) compared to unmodified CAL72, run in parallel (Fig. 7i–k). Hence, PARPi treatment elicits signalling consistent with replication checkpoint activation in RB1-defective cells, indicative that DNA replication fork impairment is a key event arising in these cells. Evidence in accord with this view is provided by fluorescence microscopy based single-cell-resolved quantification of DNA-bound DAPI (Supplementary Fig. 7i, j), documenting a significant increase of cells with >2–4n DNA following PARPi treatment, indicative of S-Phase checkpoint activation in naturally RB1-defective OHSN and RB1-normal CAL72 following RB1 ablation, but not unmodified CAL72.

**Requirement of DNA replication for PARP inhibitor toxicity in RB1-defective cells.** To address if DNA replication is a requirement for toxicity of PARP inhibition to unfold in RB1-defective osteosarcoma, we assessed whether preventing this process prevents PARPi-induced death. We cultured RB1-defective OHSN in medium containing excess thymidine to halt DNA

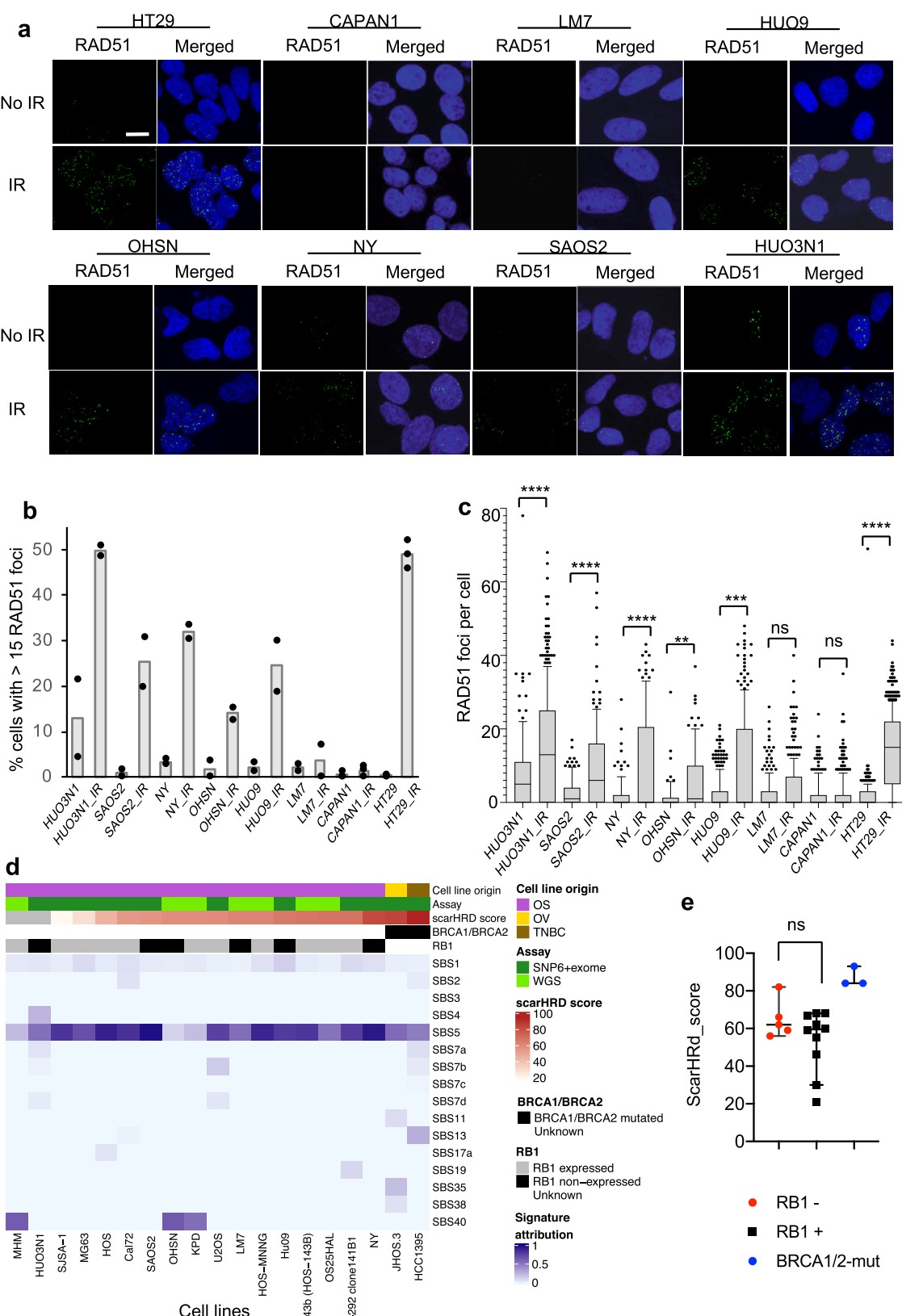

replication during olaparib treatment (Fig. 8a). Subsequently, we quantified cell death measuring SYTOX^TM death-dye uptake using time-lapse imaging. OHSN cells treated with olaparib whilst under thymidine-induced DNA replication block showed a striking,

highly significant reduction in cell death rate, compared to cycling cells. Yet death response was restored to levels similar to that in cycling cells when cells were released from the thymine-induced block prior to olaparib addition, ($^{ns}p = 0.1736$) (Fig. 8b, c). These

**Fig. 5 HR capability in RB1-defective osteosarcoma cell lines. a–c** DNA damage-dependent RAD51 recruitment in RB1-defective osteosarcoma cell lines. Cells grown on glass coverslips were irradiated (4 Gy) or left untreated. Cells were fixed 1 h following IR, subjected to immunostaining for RAD51 and nuclear foci scored using confocal microscopy. A minimum of 100 cells per line were assessed across two or more biological independent datasets. **a** Raw confocal images. Scale bar, 10 μm. RAD51 foci (green) and merged with images for DNA counterstaining using DAPI (blue). **b** Bar chart depicting quantitation of RAD51 nuclear foci. Bars depict the % of cells with >15 nuclear foci (mean ± SEM) for $n = 2$ biologically independent datasets for each cell line. **c** Box and Whiskers plot depicting RAD51 foci numbers per cell. Boxes denote 25th (min) and 75th (max) percentile, lines denote median, and whiskers 10–90 percentile. $^{ns}p > 0.05$, $^{**}p < 0.01$, $^{***}p < 0.001$, $^{****}p < 0.0001$ using a Kruskal–Wallis test with Sidak's multiple comparisons correction. HUO3N1 ($p < 0.0001$), SAOS2 ($p < 0.0001$), NY ($p < 0.0001$), OHSN ($p = 0.0065$), HUO9 ($p = 0.0001$), LM7 ($p = 0.5338$), CAPAN1 ($p = 0.9417$), HT29 ($p < 0.0001$). **d** Mutation signature analysis. scarHRD score and single-base substitution (SBS) signature extraction is detailed in Supplementary Methods. Columns are ordered by increasing scarHRD score. Cell line origin is as indicated; osteosarcoma (OS) ovarian (OV), triple-negative breast (TNBC). *BRCA1,2* and *RB1* status and the nature of sequencing information used (assay) (Affymetrix™ Genome-Wide Human SNP Array 6.0 (SNP6) + whole exome sequence(exome) or whole genome sequence (WGS), are indicated for each cell line. **e** Scatter plot comparing scarHRD scores in *RB1*-mutant (red) and *RB1*-normal (black) osteosarcoma cell lines. Error bars represent median ±95%CI. Data for cell lines with *BRCA1,2* alteration (JOSH.3 (ovarian) (*BRCA1* p.E1593*, *BRCA2*_p.R1751*), HCC1395 (breast) (*BRCA1*_p.YRRGA1202fs), and CAPAN1 (pancreas) (*BRCA2*_p.YRRGA1202fs), are included (blue). $^{ns}p = 0.33$ using a two-sided Mann–Whitney test. Source data are provided as a Source Data file.

results provide direct evidence that ongoing DNA replication is required for death to unfold in response to PARPi treatment in RB1-defective osteosarcoma.

**PARP inhibitor yields robust single-agent activity in patient-near and in vivo preclinical models of RB1-defective human osteosarcoma.** Given the substantive single-agent PARPi sensitivity of RB1-defective osteosarcoma cells in cell-based datasets we assessed whether the single-agent sensitivity extends to in vivo models of human osteosarcoma. To this end we generated xenografts of RB1-defective OHSN in immunodeficient NRG mice. Following tumour formation, mice were randomised and treated once daily for three successive 5-day periods with either vehicle or talazoparib at 0.33 mg/kg (Fig. 9a–d).

Treatment using this schedule was well tolerated, with no adverse impact on weight (Supplementary Fig. 8a) or other adverse effects observable. However, a highly significant reduction in tumour growth, apparent after a single 5-day cycle ($^{**}p < 0.01$), was seen in the talazoparib-treated compared to vehicle-treated mice. Notably, while tumours in vehicle-treated mice progressed rapidly (Fig. 9a), reaching the maximally allowable size by 22 days, none of the tumours in talazoparib-treated mice progressed to this level within that time. Importantly, and although dosing of talazoparib was discontinued on day 20, >70% of the tumours in talazoparib-treated mice remained within allowable limits at day 26, when observation was terminated (Fig. 9b). Treatment of mice carrying xenografts of RB1-normal CAL72, using an identical schedule, failed to show reduction in tumour growth compared to vehicle-treated controls, and revealed disease progressing indistinguishable from that of the placebo control arm (Fig. 9c, d and Supplementary Fig. 8b). While detailed studies using isotype-matched cell lines with engineered RB1 loss would be necessary for proof of-concept of RB1 dependence, the results suffice to document single-agent effects in agreement with those observed in cell-based experiments, and coherent with such an assumption.

We further assessed the response of PARPi in three different early-passage patient-derived models of metastatic osteosarcoma adapted for growth in 2D tissue culture, two of them positive for RB1 expression and one without detectable RB1 (Fig. 9k). Recent work documents gene expression patterns highly correlative with those of their respective xenograft or primary tumour material indicative of their phenotypic resemblance of the disease they are derived from[45]. Treatment using either PARP inhibitor olaparib (Fig. 9e–g) or talazoparib (Fig. 9h–j) revealed a significantly higher dose sensitivity of the RB1-defective PDX (PDX-OS19-C2) assessed using clonogenic survival compared to either of the

RB1-normal lines (PDX-OS25-C1 and PDX-OS16-C2), with IC50 values in the submicromolar range for olaparib (Fig. 9g) and the low nanomolar range for talazoparib (Fig. 9j), in line with values obtained for the RB1-defective group of established osteosarcoma cell lines analysed earlier.

These data provide evidence that the single-agent PAPRi sensitivity observed in cell-based assay translates into substantive single-agent preclinical anti-tumour activity, yielding reduced disease progression and extended survival in mice carrying human RB1-defective osteosarcoma xenografts and selective sensitivity in patient-near models.

## Discussion
Our work identifies PARP1,2 inhibition as a synthetic vulnerability and therapeutic opportunity for *RB1*-mutated osteosarcoma with additional evidence that deleterious *RB1* mutation may be a biomarker for clinically relevant PARP1,2 inhibitor sensitivity in other cancers. PARPi are in current clinical use, with notable effect on quality of life and overall survival in multiple cancers[46]. Their existing clinical utility highlights the imminent opportunity for clinical translation of the results we report here.

Our work documents that enforced RB1 loss causes clinically meaningful sensitivity (i.e. sensitivity akin to that seen in *BRCA1,2*-defective CAPAN1) in otherwise PARPi insensitive osteosarcoma lines, providing proof of concept for a direct role of RB1 loss in the selective PARPi sensitivity observed. The lack of evidence for frank HRd in cancer cell lines with RB1 loss raises questions as to the mechanism that underlies their inhibitor sensitivity. Our work positively identifies PARPi trapping and active DNA replication as mechanistic prerequisites for sensitivity, paralleling observations in cancers with HRd[26,41]. These observed similarities argue for a shared inability of cancers with HRd or *RB1* loss to avert the lethal consequence of trapped PARP complexes, which is known to underlie PARPi inhibitor sensitivity caused by HRd.

Published data propose a role of RB1 in HR, entailing E2F1-dependent recruitment of chromatin remodelling activity to sites of DNA damage[18], albeit, the scale of HRd arising through this mechanism has not been assessed. It is conceivable that localised HRd arising within subgenomic contexts, while not detected in genome-wide mutation spectrum analysis or identified by principal incompetence to recruit RAD51, could cause the PARPi hypersensitivity observed.

Intriguingly, a recent functional genomics screen identified HRd as selectively lethal in cell lines derived from retinoblastoma, a tumour primarily initiated by RB1 loss. While it is not yet clear whether a similar selective lethality exists in other cancer types

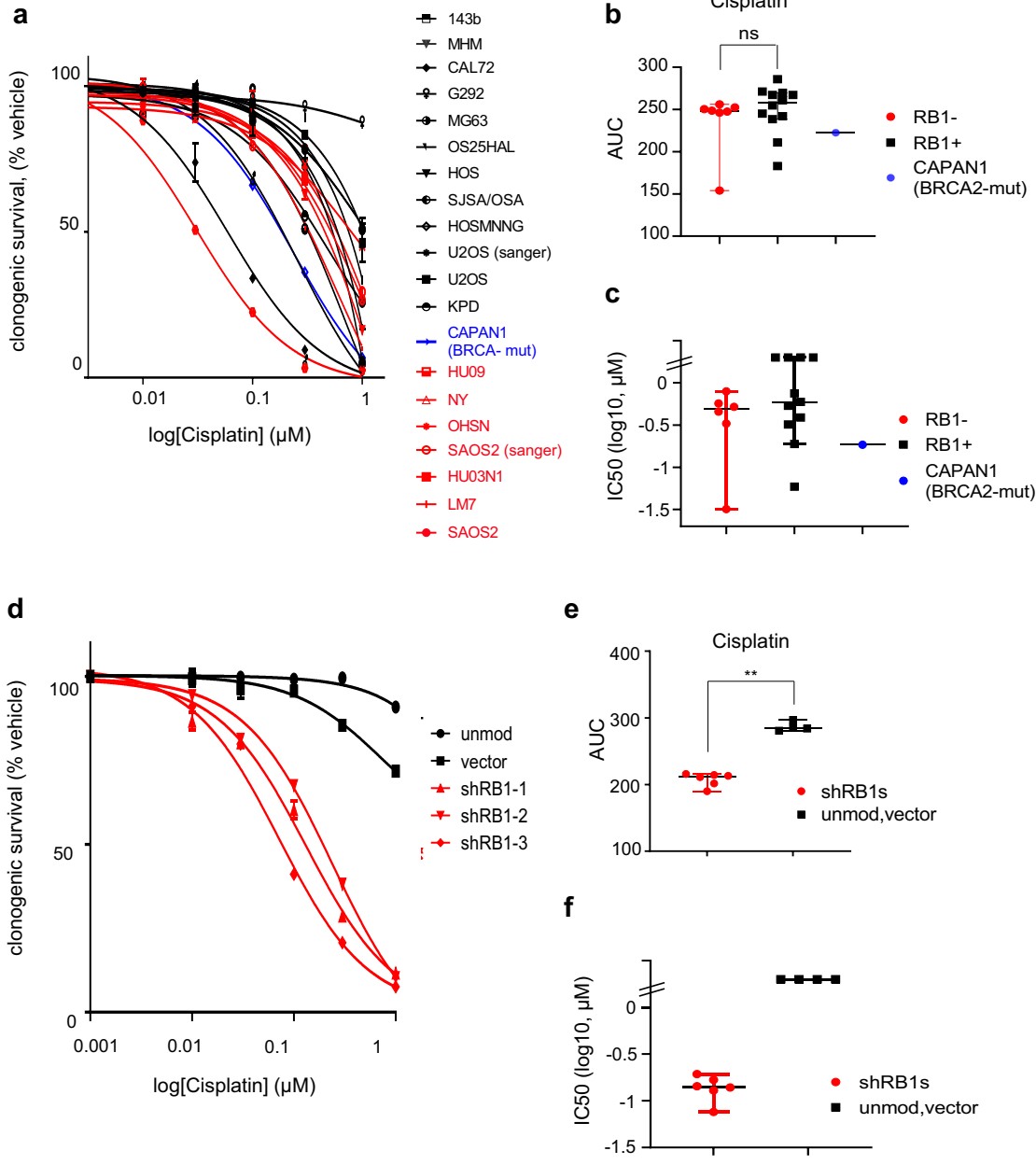

**Fig. 6 Platinum sensitivity in RB1-defective osteosarcoma.** Cells seeded at low density into six-well plates were cultured in the presence of increasing concentrations of cisplatin or vehicle (DMSO). Colonies arising were stained using crystal violet dye. Clonogenic survival was quantified using dye extraction. **a–c** Platinum response in RB1-defective (red) or RB1-normal (black) osteosarcoma and *BRCA2*-mutant CAPAN1 (blue). **a** Concentration-response curve and **b** scatter plots depicting AUC value comparison and **c** log IC50 values, deduced from the concentration-response data in **a**. Bars in scatter plots (**b**, **c**) depict median (±95%CI), calculated using a two-tailed Mann–Whitney test, *p* (**b**) = 0.3730ns. Data points in **a–c** reflect the mean of parallel duplicate wells for *n* = 1 dataset. **d–f** RB1-normal osteosarcoma CAL72 cells transduced with lentivirus vector encoding different *RB1*-targeting shRNAs (shRB1-1, shRB1-2 or shRB1-3) (red), or empty vector backbone (vector), or left unmodified (unmod) (black). **d** Concentration-response curve, **e** scatter plots depicting AUC value comparison and **f** log IC50 values, deduced from the concentration-response data in **d**. Bars in scatter plots (**e**, **f**) depict median ±95%CI. Data in **d** depict means of parallel duplicate wells for one of *n* = 2 biologically independent datasets, data in **e** and **f** summarise results for *n* = 2 biologically independent datasets. *p* (**e**) = 0.0095, calculated using a two-tailed Mann–Whitney test, **p < 0.01. Source data are provided as a Source Data file.

with RB1 loss, the results as they stand could suggest that frank HRd is not acceptable in *RB1*-mutated backgrounds[47].

Patient selection in current clinical applications relies on evidence of HRd in cancer tissue[48,49]. However, genomic or functional evidence for frank HRd is not detectable or significantly associated with *RB1* loss in osteosarcoma. This observation raises the case that patients carrying such cancers while potentially

benefiting from PARPi are not identified using the currently clinically approved criteria for PARPi use.

An increasing number of reports link mechanisms unrelated to HRd to single-agent PARPi hypersensitivity, reviewed in ref. [43]. These mechanism include defective DNA cohesion and chromatin remodelling[50,51], defective transcription-coupled repair[52], defects in replication fork stabilisation[53], defects in S-phase

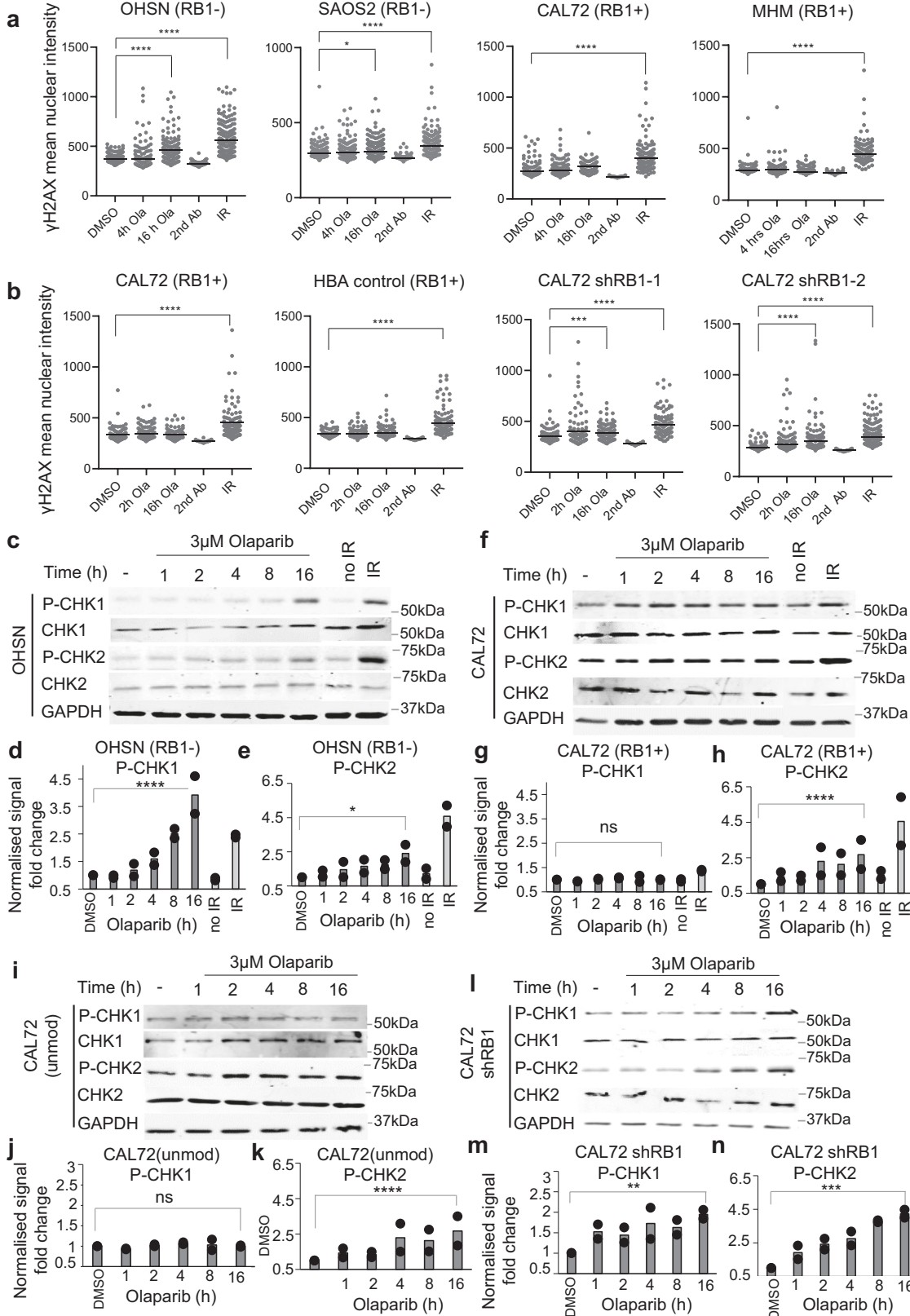

checkpoint functions[54], events that instigate replication stress[55], events that increase single-strand DNA breakage[56], and loss of PARP1-dependent gene transcription[57]. Defects in serval of these processes, including DNA cohesion and chromatin remodelling[58], and metabolic changes leading to replication stress[59] are known to result from RB1 loss, which in turn could explain the observed sensitivity phenotype.

While our work advocates the use of PARPi in *RB1*-mutated osteosarcoma, comprehensive preclinical validation, including how PARPi would be best integrated into the current

**Fig. 7 Effect of PARP inhibition on DNA damage response in cells with different RB1 status.** DDSB repair signalling assessed using anti-phospho (Ser139) H2AX (γH2AX) immunofluorescence, **a** in osteosarcoma cells with different RB1 status, or **b** in RB1-normal osteosarcoma CAL72 transduced with lentivirus vector encoding *RB1*- or irrelevant control haemoglobin (HBA)-targeting shRNA. Cell lines after treatment with DMSO, 3 μM olaparib, or 2.5 Gy of IR. Cells were exposed to olaparib for 2 or 4 or 16 h or allowed to recover for 1 h after IR. Scatter plots report data distribution and mean for parallel duplicate samples from one of $n = 2$ biologically independent datasets, with statistical assessment using a two-tailed Mann–Whitney test. **c–n** DDSB repair checkpoint signalling assessed using phospho(Ser345)-CHK1 (pCHK1) and phospho(Thr68)-CHK2 (pCHK2) quantitative immunoblot analysis. **c–e** RB1-defective OHSN, **f–h** RB1-normal osteosarcoma CAL72 and **i–n** RB1-normal CAL72 without or with shRNA-mediated *RB1* depletion. Cell lines after treatment with vehicle, 3 μM olaparib, or 2.5 Gy IR. Cells were exposed to olaparib for 1, 2, 4, 8 or 16 h or IR with recovery for 1 h. GAPDH was used as loading control. CHK1 and CHK2 denote immunoblot signals for pan-CHK1 and pan-CHK2. Representative raw data blots (**c**, **f**, **i**, **l**) or bar graphs (**d**, **e**, **g**, **h**, **j**, **k**, **m**, **n**) depicting fold change of pCHK1/2 signal relative to vehicle-treated cells after normalisation to GAPDH. Bars summarising mean of $n = 2$ biologically independent datasets, with statistical assessment of median standardised raw values using a two-sided one-way ANOVA test. $p$ (DMSO vs Ola 16 h) OHSN (pCHK1) < 0.0001****, OHSN (pCHK2) = 0.0107*, CAL72 (pCHK1) = 0.9937ns, CAL72 (pCHK2) < 0.0001****, CAL72 ShRB1 (pCHK1) = 0.0049**, CAL72 ShRB1 (pCHK2) = 0.0001***. Source data are provided as a Source Data file.

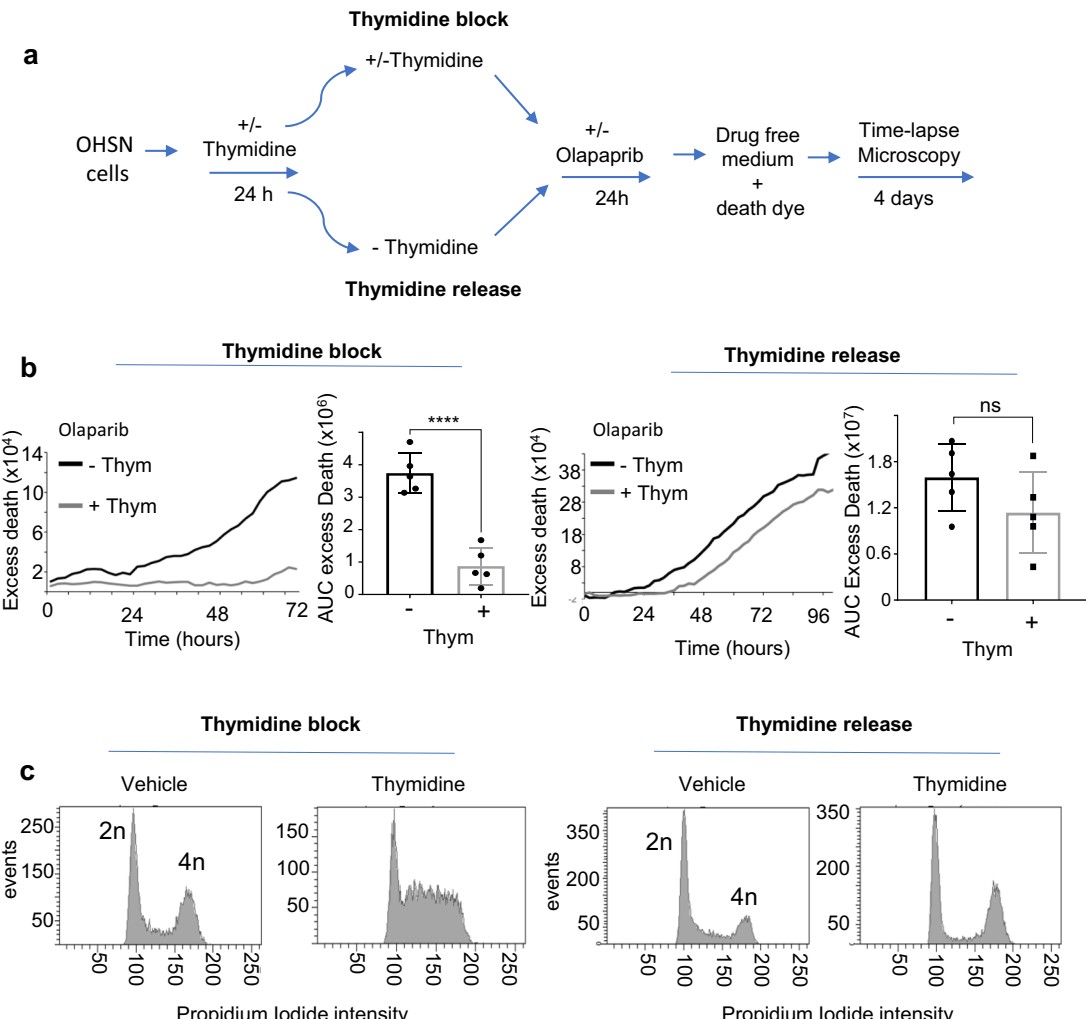

**Fig. 8 DNA replication impairment rescues PARPi sensitivity. a–c** PARPi sensitivity following DNA replication perturbation. **a** Experiment design for assessing the role of DNA replication in PARPi sensitivity. RB1-defective OHSN seeded in 12-well plates were treated as indicated, then subjected to time-lapse microscopy in the presence of SYTOX™ death-dye. **b** Death assessed by SYTOX™ death-dye incorporation. Raw traces depicting excess death above vehicle for one representative experiment, and bar graphs depicting AUC values summarising excess death over vehicle (±SEM) for $n = 5$ biologically independent datasets, $p$ (thymidine block ($-$ thym vs $+$ thym)) < 0.0001****, $p$ (thymidine release ($-$ thym vs $+$ thym) = 0.1736ns, calculated using unpaired, two-tailed Student's $t$ test. **c** Exemplary cell cycle profiles documenting the effect of thymidine treatment on cell cycle progression. Cells seeded in parallel six-well plates were treated as for **b**, then analysed using flow cytometry at 32 h. Data shown are from one representative dataset of $n = 4$ biologically independent datasets. Source data are provided as a Source Data file.

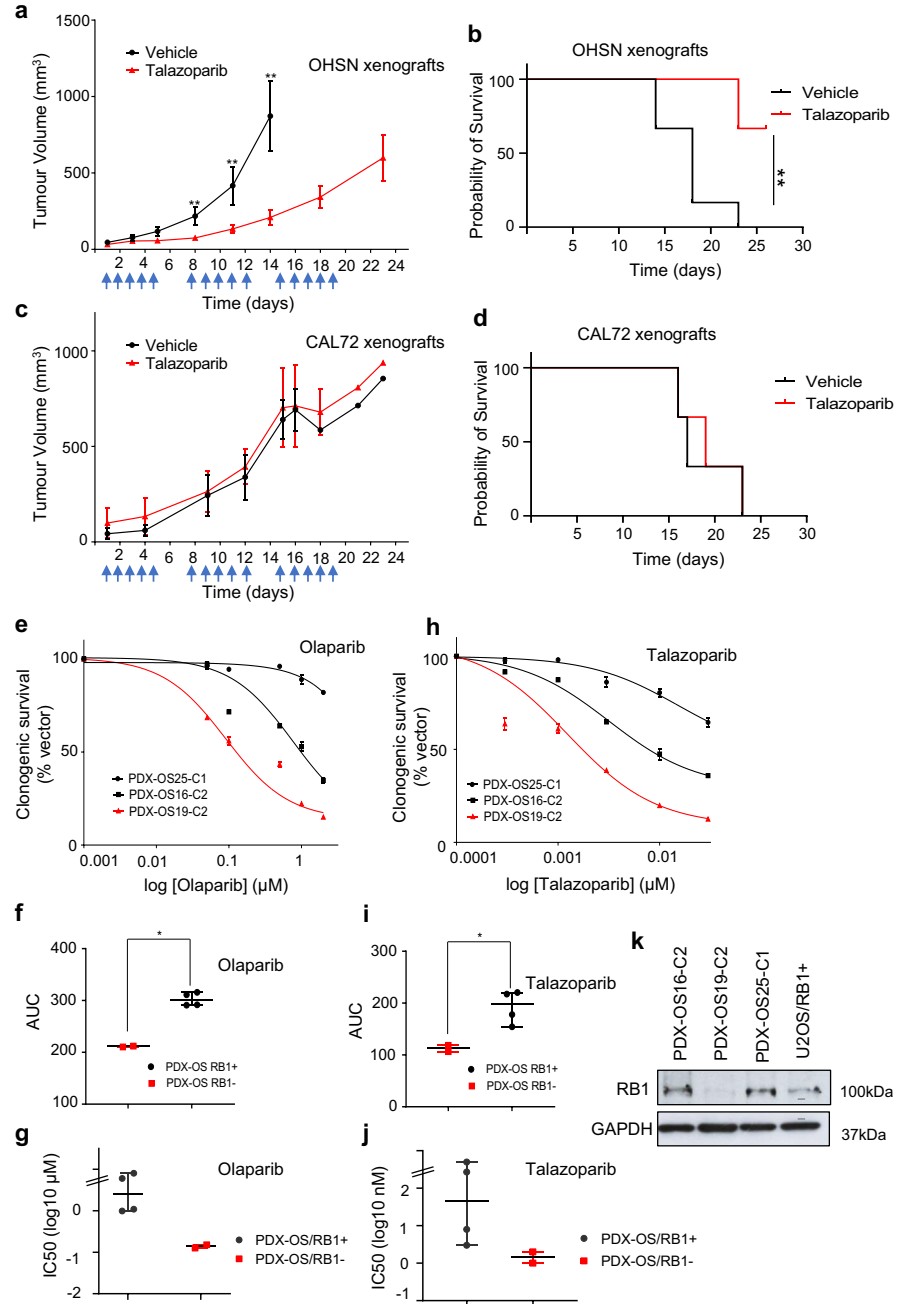

**Fig. 9 Selective single-agent PARPi sensitivity in in vivo xenograft and patient-near 2D models. a–d** Tumour response to single-agent PARPi treatment. NRG (NOD.Cg-Rag1tm1Mom Il2rgtm1Wjl/Szj) mice carrying OHSN or CAL72 tumour xenografts were treated daily (five times per week) with the PARPi talazoparib at 0.33 mg/kg, or vehicle for 3 weeks. **a, c** Tumour volumes over time, measured at the indicated time points. Arrows indicate dosing schedule of PARPi or vehicle. Data points represent mean ± SEM for $n = 6$ (OHSN) or $n = 3$ (CAL72) mice per treatment arm. $p$ (**a**) day 8 = 0.0087**, day 11 $p = 0.009$** and day 14 $p = 0.0022$** calculated using a two-sided unpaired Student's $t$ test, $p$ (**c**) day 8 = 0.7$^{ns}$, day 11 $p > 0.99^{ns}$ and day 14 $p > 0.99^{ns}$ calculated using a two-sided unpaired Student's $t$ test. **b, d** Kaplan–Meier survival analysis of mice assessed in **a** and **c**. $p$ (**b**) = 0.0027**, $p$ (**d**) = 0.8584$^{ns}$, calculated using a log rank (Mantel–Cox) test. **e–j** PARP inhibitor response in patient-near 2D models assessed using clonogenic survival. Concentration-response curves (**e**, **h**), depicting results for RB1-defective (red) or RB1-normal (black) patient-near osteosarcoma models after treatment with olaparib, or talazoparib. Data reflect the mean ± SD of parallel duplicate wells for one of $n = 2$ biologically independent datasets. Scatter plots (**f, i**) summarising AUC values from $n = 2$ biologically independent datasets involving olaparib, $p$ (f) = 0.012*, or talazoparib, $p$ (**i**) = 0.0114*, calculated using a two-tailed Mann–Whitney test. Bars depict median and ±95% CI. Scatter plots (**g, j**) summarising IC50 values deduced from dose response data from $n = 2$ biologically independent datasets. Bars depict median ±95% CI. Immunoblot analysis (**k**) assessing the expression of RB1 in osteosarcoma patient-near cell models. GAPDH was used as loading control. Source data are provided as a Source Data file.

management of osteosarcoma, will be of likely paramount importance to ensure clinical benefit.

PARPi are rapidly moving into first-line clinical use in patients with HRd ovarian cancers and considerable efforts are underway to extend their use to other cancers. Most pertinent to the work reported here is the planned assessment of PARPi within the paediatric MATCH study (NCT03233204), a large-scale precision medicine trial in children, adolescents, and young adults with advanced cancers including osteosarcoma, with use of *BRCA1,2* mutation or HRd for patient selection. The highly penetrant hypersensitivity in RB1-defective osteosarcoma cells shown here combined with the currently limited options in patient with such cancers, advocates expansion of assessment to include *RB1*-mutated disease.

## Methods

**Cell lines, chemicals and antibodies**. The osteosarcoma tumour cell lines and the primary cultures PDX-OS16-C2, PDX-OS19-C2 and PDX-OS25-C1 were described previously[15,60,61]. PARPi and cisplatin were purchased from Selleck Chemicals. Antibodies and shRNAs are detailed in Supplementary Materials. Mutation spectrum analysis was as described[39].

**Assessment of drug response**. Drug sensitivity was assessed in 96-well plates based on resazurin-reduction 5 days following drug addition. Clonogenic survival assessments and immunofluorescence staining were performed as described[62]. Time-lapse microscopy was performed in 96-well plates as described in ref. [63] using an IncuCyte ZOOM live cell analysis system (Essen Bioscience). For cell cycle analysis, cells were fixed in 70% ethanol, stained using propidium iodide and analysed using flow cytometry. Immunoblot analysis used whole-cell protein extracts prepared by lysis of cells into 0.1% SDS, 50 mM TRIS-HCL, pH 6.8, containing protease and phosphatase inhibitors (Thermo Fisher Scientific, UK). In vivo experiments were carried out under UK Home Office regulations in accordance with the Animals (Scientific Procedures) Act 1986 and according to United Kingdom Coordinating Committee on Cancer Research guidelines for animal experimentation[64] with University College London's Animal Welfare Ethical Review Body (AWERB) approval. Tumour growth was assessed twice weekly using digital callipers. Assessments were terminated in accordance with AWERB guidelines.

**Statistical analysis**. Statistical hypothesis testing was performed using Microsoft Excel or GraphPad Prism. Statistical tests used are named within the text. Differences with $p < 0.05$ were considered statistically significant.

Comprehensive method and material information are provided under Supplementary Materials.

**Reporting summary**. Further information on research design is available in the Nature Research Reporting Summary linked to this article.

## Data availability

The data supporting the findings of this study are available within this article, the Supplementary Information or from the authors upon reasonable requests. Source data are provided with this paper. Sequencing data used in this study are available in the European Nucleotide Archive, hosted by EMBL-EBI (https://www.ebi.ac.uk/ena/browser/home) under study accession code PRJEB47683. Source Data are provided with this paper.

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

## Acknowledgements

G.Z., C.A.-M. and M.D. were supported by Children with Cancer UK (ref. 17-244, PI; S.M.). The research leading to these results has received funding from AIRC under IG2016-ID. 2018451 project, PI; K.S. and IG2019-ID. 22805 project, PI; K.S., with additional support through philanthropic giving by charities aid foundation account A80105053 (PI; S.M.). The Francis Crick Institute receives its core funding from Cancer Research UK (FC001202), the UK Medical Research Council (FC001202) and the Wellcome Trust (FC001202). M.T. was supported as a postdoctoral researcher of the F.R.S.-FNRS. C.M. was supported by the CRUK UCL Centre Non-Clinical Training Award [C416/A23233]. M.T. was supported as a postdoctoral researcher of the F.R.S.-FNRS. S.J.S. was funded in part by NIHR UCLH Biomedical Research Centre.

## Author contributions

G.Z., N.P., S.J.S., K.S. and S.M. contributed to the design of datasets and strategy; S.M. lead the project; G.Z. and S.M. wrote the manuscript; G.Z., C.A.-M., C.M., R.-M.A., M.D., E.R., L.R.D. and C.C. performed components of the experimental work and analysed associated data; C.D.S. and M.T. performed the bioinformatics-based genomic scar analysis and J.M. assisted with microscopy and high-content data analysis.

## Competing interests

The authors declare no competing interests.
