## [Peer Review File · Nature Communications]

Reviewers' Comments:

Reviewer #1:

Remarks to the Author:

The authors have produced a mechanistic and translational dive into the potential utility of PARPi in osteosarcoma, finding that it is more efficacious against RB1-mutant or -lost osteosarcomas, through a replication fork stalling mechanism.

The noted correlations in Figure 1g/h are troubling on two counts. First, the correlation of olaparib with itself is lower than with other drugs. That makes how good the other drugs appear to be suspicious. Second, if the argument that the particular mechanism of sensitivity is RB1-loss dependent, why are the RB1+ cell lines also showing such strong correlations between the varied PARPi.

The most compelling experiments were those performed with knock-down of the tumor-suppressor RB1 in otherwise RB1-retaining cells, showing increased sensitivity to cisplatin or PARPi. The major limitation was that these were typically done with only a single cell line, each, even if with 2 or 3 different shRNAs. Adding even a second example of cell line would greatly strengthen the findings.

Attention to rigor could be improved with many n=1 or n=2 experiments, and notations that one is shown of 2 or more repetitions. How many were actually done? For quantitative westerns, you could just do them 3 times and report the results with error bars and statistics.

The single experiment with the xenografted cell line shows a fairly modest effect. Why were the mice culled after day 26? Could they have survived longer? Also, what was the survival analysis really based on, only size below a certain threshold?

Reviewer #2:

Remarks to the Author:

PARP Inhibitor RB1 Mutant Osteosarcoma Nature Comm

This manuscript shows that osteosarcoma cell lines with RB1 mutations are more sensitive to PARP inhibitors. It also demonstrates that siRNA knockdown of RB1 sensitizes RB wild-type cells to PARP inhibitors and cisplatin. Surprisingly, despite the sensitivity to PARP inhibitors, there was no defect in RAD51 foci formation, a key step in the homologous recombination repair pathway. Additional studies characterized the DNA damage responses that are induced by PARP inhibitors in RB1-mutated cells, demonstrated that cells must replicate DNA to be sensitive to PARP inhibitors, and showed that the growth of a xenografted RB1-mutated cell line tumor was slowed by a PARP inhibitor.

The experiments are, for the most part, well done and the results look solid. Importantly, the results suggest that PARP inhibitor may have activity in RB1-mutated tumors, which are common in osteosarcoma. There are multiple concerns, however.

- 1) There is no insight into mechanism. Most of the data are descriptive and only characterize survival and other phenotypic differences.
- 2) The studies do not effectively rule out an HR defect. The studies only show that there isn't a RAD51 foci formation defect. This does not mean that there isn't a defect in homologous recombination. Moreover, even if the studies had demonstrated an HR defect in RB1-mutated cells, there still would be no mechanistic insight
- 3) The fact that RB1 knockdown sensitizes to cisplatin and PARP inhibitor suggests that there is an HR defect at some level (although not at the level of RAD51 deposition).
- 4) The fact that there is no correlation of cisplatin sensitivity with RB1 status in the collection of cell lines does not necessarily support a lack of an HR defect. These differences are likely due to the fact that a large number of factors affect cisplatin sensitivity. Therefore, it is not surprising that these cell lines would have greatly different sensitivities to platinum. Additionally, some of the

cell lines may have been derived from platinum-resistant tumors.

5) The studies do not show that an RB1-mutation (or deficiency) actually affects the antitumor activity of talazoparib in the xenograft model. Instead, they show that the xenograft grows more slowly – but does not shrink – when treated with talazoparib. This could be true for many tumors, even those that lack known HR defects.

6) Similar findings have been previously reported. A previous publication “Osteosarcoma cells with genetic signatures of BRCAness are susceptible to the PARP inhibitor talazoparib alone or in combination with chemotherapeutics” (PMC5564725) concluded that an HRD-LOH signature occurred in some osteosarcoma cell lines and was associated with sensitivity to talazoparib. These findings, which are somewhat contradictory to findings reported here, were not discussed.

Minor comments:

1) The use of the word “matched” to describe cell lines is potentially misleading. The cell lines that were being compared were from different patients. Depending on your viewpoint, “matched” could mean that the cell lines being compared were quite similar (e.g., a mutation in a specific gene).

2) In many figures where a representative experiment is shown, it’s not clear how many times the experiment was repeated with a similar result.

Reviewer #3:

Remarks to the Author:

This interesting paper describes sensitivity of multiple RB1 null cancer lines to PARP inhibitors (PARPi) relative to RB1 wild type lines, with a subsequent focus on osteosarcoma. This contrast was consistent across various PARPi inhibitors, and RB1 null osteosarcoma cells showed similar median sensitivity to these drugs as a BRCA2 null pancreatic cancer line. The cellular mechanism involved rapid death. Depleting RB1 in a single osteosarcoma cell line (CAL72) increased sensitivity to PARPi. Sensitivity of BRCA mutant cells to PARPi is linked to defective homologous recombination (HRd), reflected in a failure to recruit RAD51 to double strand breaks (DSBs). However, in most RB1 null osteosarcoma lines, there was RAD51 recruitment after radiation-induced DSBs. Mutation spectrum analysis also argued against any deficit in HR machinery in RB1 null osteosarcoma. Moreover, BRCAness typically predicts sensitivity to platinum drugs, but RB1 absence did not increase sensitivity to cisplatin (although, surprisingly, depleting RB1 in CAL72 cells did). Although HR appeared intact (at least according to RAD51 recruitment), PARPi induced more DSBs in RB1 null vs wt osteosarcoma lines, as assessed by H2AX staining, and DSB induced checkpoint, as indicated by increased pCHK1. Thymidine block assays showed that PARPi induced cell death requires cell cycle progression. Xenograft assays with one RB1 null line indicated a positive therapeutic effect.

Comments

The identification of RB1 loss as a sensitizing event to PARPi will be important to the cancer field. The fact this sensitivity is due to something other than HR deficiency (as in BRCA mutant tumors), is also of note, although the underlying mechanism remains to be determined. The paper is well written, the experiments are well done, and there is broad clinical relevance. There are some areas of concern, particularly in terms of the number of cell lines used to directly address the functional relevance of RB1 absence.

1. The correlation between different RB1 wt or null lines is striking, but a direct role for RB1 was tested in only one cell line (CAL72). Is it possible to add another line for at least some of the assays to add extra confidence that RB1 loss enhances sensitivity? This seems relevant given that depleting RB1 in CAL72 cells increased sensitivity to platinum, which was not reflected in comparison of other RB1 null or wt osteosarcoma cells. Perhaps CAL72 is also unusual in its response to PARPi after RB1 depletion? The correlations suggest otherwise, but it would be reassuring to show this more definitively.

2. On a related note, is it feasible to ask whether adding RB1 back to RB1 null cells (to levels

similar to those in RB1 wt osteo) reduces sensitivity? This would not need to be done in a large number of lines (one or two would suffice), but would add confidence (to the depletion assays above) that RB is the critical determinant. It may be difficult/impossible to create stable lines with RB added back, but might be feasible to use viral vectors to establish a population of transduced cells for the drug study. The authors have done a very nice job correlating RB1 loss to PARP sensitivity, and it would add so much to their paper to confirm the functional connection.

3. If cell survival with PARPi does, as predicted, fall with RB1 depletion and rise with RB1 restitution, it'd be important to test whether this also reflects difference in DSB induction.

4. The in vivo work is also somewhat limited, again based on a single line, and no comparison of isotype lines +/- RB1. I wouldn't insist on more of these tumor assays given the work required, but the authors should at least acknowledge the serious limitations of their analysis, and note that more rigorous studies are required to test sensitivity of RB1 null vs wt tumors to PARPi in vivo.

5. The Discussion touches on the puzzle as to why PARP inhibitors are so effective in RB1 null osteosarcoma, despite the fact that HR appears to be intact. It could be a bit better developed to summarize different theories as to how PARP inhibitors kill cells, which are nicely summarized by Rose et al (PMID: 33015058). Whether any of these mechanisms is responsible in RB null osteosarcoma is beyond the scope of this paper, but it does seem critical to discuss in more depth. It is surprising that HR-intact cells are so sensitive. In that vein, it was reported recently that another RB1 null cancer, retinoblastoma, also has intact HR, but is hypersensitive to HR inhibition, and a RAD51 inhibitor synergized with agents like topotecan which, like PARP perturb replication progression (Aubry et al, PMID: 32572160). Perhaps all RB1 null cells are hyper-dependent on HR, and that dependency could contribute to PARP sensitivity? The authors could explore the possibility that PARP synergizes with HR inhibitors, at least as a Discussion item.

Typos

Fig 1 a/c/e legend - "back" should be "black".

Fig 8e graph "Vehcile" = Vehicle

Reviewer 1

We thank the reviewer for commenting on our manuscript. The reviewer raised several specific issues.

Correlations presented in Figure 1g/h are troubling because the correlation of olaparib with itself is lower than with other drugs.

Pearson's correlation coefficient is not a quantitative statement of likeness (and should not be used as such). It is a statistical/ probabilistic measure reporting strength of a linear relationship between paired data.

The correlations that we measure in all instances is very strong (>80%). They support the claims that we make in the text. Namely within the same drug they provide evidence for highly significant "likeness" concerning data reproducibility, between drugs they provide evidence for statistically significant "likeness" concerning the mechanism on which action is based.

Why are the RB1+ cell lines also showing such strong correlations between the varied PARPi.

Our raw data (Figure 1a-e) document that there is response in a number of RB+ cells when high doses of inhibitor are used. In most cases 10 to 100-fold higher doses are needed to give a similar effect size. High effect size (i.e. 100% inhibition) cannot be reached in the majority of RB+ lines even if concentrations approaching solubility limits of drugs are used.

With Pearson testing allowing statistical analysis of response, the response observed can be assessed even if it is poor. Our data show that a highly significant correlation exists, suggesting that the response seen (be it poor) is caused by an activity shared between the different inhibitors (and hence by definition qualifies as being due to "on-target" inhibitor action). The analysis is not central to the argument or message of the paper, but we felt that for completeness sake and in adherence to good practice, it would be valuable to include it.

The most compelling experiments were those performed with knock-down of the tumor-suppressor RB1 in otherwise RB1-retaining cells, showing increased sensitivity, but the major limitation was that these were typically done with only a single cell line, each, even if with 2 or 3 different shRNAs. Adding even a second example of cell line would greatly strengthen the findings.

We are now adding data for a second cell line, 143b, reporting on a) key phenotypic consequence, namely selective decrease in day-5 viability (**shown in Supplementary Figure 4 e-g**) and b) key mechanistic consequence, namely the PARPi inflicted increased in gammaH2AX positivity (**shown in Supplementary Figure 7d to h**). Both sored significantly altered upon RB1 ablation, as we had shown for RB1-ablated Cal72, and in line with predictions based on naturally RB1-mutated and RB1-normal osteosarcoma lines.

Attention to rigor could be improved with many n=1 or n=2 experiments, and notations that one is shown of 2 or more repetitions. How many were actually done?

With apologies, and have revised, identifying more accurately the number of repeats that each figures component represents. Revised statements are underlined in the resubmitted text.

All conclusions are based on a minimum of two independent datasets and key conclusions are supported by orthogonal assays, for example day-5 viability and clonogenic activity.

A higher number of repeats are available for specific cell lines, because we used them to benchmark data consistency where experiments needed to be broken up into batches, or because these lines were co-analysed to benchmark outcome in other experiment settings. For example, we have four datasets for day-5 response curves for CAL72 and 143b because these lines were co-analysed alongside the shRNA knockdowns and similar applies for OHSN and SAOS2.

For quantitative westerns, you could just do them 3 times and report the results with error bars and statistics.

We now present summary data for two independent experiments across all conditions, shown as bar graphs +/- STD in **Figure 7d-h and k-o**, with analysis-of-variance (ANOVA) statistical statements. We also include new evidence based on cell cycle analysis, which provides orthogonal/ independent documentation for prominent S/G2 phase checkpoint activation in RB1-defective contexts (data shown in **Supplementary Figure 7i, k**)

The single experiment with the xenografted cell line shows a fairly modest effect.

The data shown document significant single-agent pre-clinical *in vivo* activity, providing a crude but critical milestone for attempts of clinical translation and indicative of potential clinical benefit.

As the reviewer is probably aware, significant *in vivo* pre-clinical efficacy does not always arise, albeit response may be seen in tissue culture models, often because cancer behaviours in monoculture 2D settings do not unfold under 3D *in vivo* tumour growth conditions. A prominent example is the single-agent hypersensitivity of Ewing Sarcoma to PARPi which does not confirm in preclinical *in vivo* experiments, potentially because the driver of sensitivity (EWS translocation), is dispensable for tumour progression *in vivo*.

The reviewer may also be aware that PARPi serve as maintenance therapy, used in patients that are principally disease free, acting by preventing distant recurrence. These inhibitors would not be effective in a neoadjuvant setting or “cure” patients featuring bulk tumour mass. The response to PARPi seen in our work is in line with their known action in clinical applications and with data seen in other preclinical models.

Why were the mice culled after day 26? Could they have survived longer? Also, what was the survival analysis really based on, only size below a certain threshold?

The experiments shown are governed by the UK home office licence under which the work was performed. We are required to euthanise mice once tumours have reached a specified size, or affect animal welfare. We confirm that loss of survival events in the data shown reflect xenografts reaching the licensed endpoint of tumour size and have stated this now more overtly in the relevant method section. Under UK/ EU regulations it is not acceptable to run experiments with death as the experimental endpoint.

Further conditions in the licence is that assessments need to be ended once the purpose of the experiment has been reached, so to limit animal welfare burden. The purpose of the experiment was to document preclinical *in vivo* single-agent efficacy, using an allowable maximal number of drug administrations (maximally 15). In adherence to these conditions the experiment was terminated once all animals in the control group had reached maximal allowable size.

We have more clearly described these licence conditions in the relevant method section

Reviewer 2

We thank the reviewer for their thought and comments.

We agree with the reviewer that the value of this publication is in translational biology, namely the **results suggest that PARP inhibitor may have activity in RB1-mutated tumors, which are common in osteosarcoma.**

There is no insight into mechanism. Most of the data are descriptive and only characterize survival and other phenotypic differences.

As stated by the reviewer the value of this publication is the identification of RB1 loss as a novel biomarker of PARPi sensitivity. We strongly focused the work at maximizing confidence to this effect. We present a novel and potentially expandable concept, whereby RB1 may serve as a biomarker predictive of PARPi response in RB1 defective cancers. Importantly we highlight novel therapeutic opportunity in a tumour type (osteosarcoma) where no mechanism based/ genome informed treatments are approved currently. We believe that our report will add compelling information which could be key in facilitating the clinical use of PARP inhibitors in osteosarcoma.

Our work sets out a framework on which mechanistic predictions are possible, e.g. that inhibitor effects size relies on PARP trapping and is dependent on S phase progression, thereby excluding transcriptional effects and effects linked to a role of BER and NER loss consequent to PARP inhibition. Most importantly we document that canonical and clinically used measures of BRCAness/ HRd would not detect sensitivity caused by RB1 loss, again an important aspect for the clinical translation of PARPi use in osteosarcoma.

We have reworded relevant section in the discussion to frame virtue and the mechanistic predictions born from the data more clearly.

The studies do not effectively rule out an HR defect. The studies only show that there isn't a RAD51 foci formation defect. This does not mean that there isn't a defect in homologous recombination. Moreover, even if the studies had demonstrated an HR defect in RB1-mutated cells, there still would be no mechanistic insight

and

The fact that RB1 knockdown sensitizes to cisplatin and PARP inhibitor suggests that there is an HR defect at some level (although not at the level of RAD51 deposition).

We are now including further “genomic scar” analysis, covering the entire cell line panel, enabled by newly established whole genome sequencing of seven cell lines for which prior sequence was unavailable. We use ScarHRd scoring, which is methodologically aligned with the FDA approved clinical biomarker strategy for identification of HRd in tumours and patients with benefit from PARPi in ovarian cancer. The results indicate that there is no significant difference in ScarHRd score levels between RB1-defective and RB1-normal osteosarcoma lines. The results are shown in **Figure 5d, e.**

We are also including additional data assessing the effect of 53BP1 ablation, which broadly abolishes PARPi sensitivity consequent to HRd, likely because 53BP1 loss prevents inappropriate use of NHEJ-based repair, leading to chromosome translocation and considered the key toxic event following from trapped PARP in cells with HRd. Our results show that 53BP1 loss does not alter PARPi sensitivity albeit leading to overt resistance in a BRCA1-

mutated breast cancer cell line, used as a control. The data are shown in **Supplemental Figure 5 b-o** and provide further evidence for a likely HRd- independent mechanisms leading to PARPi sensitivity in RB1-defective cells.

Naturally, it is never possible to categorically exclude that some form of HRd arises, and we acknowledge this in our discussion. Our data make a strong case that if there was HRd it does not align with canonical features in terms of genomic “tell-tail” signatures and RAD51 recruitment, and cannot be identified using canonical or clinically used strategies for HRd identification.

The fact that RB1 knockdown sensitizes to cisplatin and PARP inhibitor suggests that there is an HR defect at some level (although not at the level of RAD51 deposition).

We agree with the reviewer that there is a tight correlation between HRd and platinum sensitivity and that it is tempting to use this correlation as evidence for HRd. However, there are other events, e.g. reduced replication fork stability and defective fork remodelling, that enhance both Platinum and PARPi sensitivity. As stated above, and in our discussion, we do not find evidence for genome-wide HRd, but these measures do not exclude the presence of localised/ regional weakness in using HRd, which might be sufficient to yield the different sensitivity phenotype observed. That said, the fact that a mechanism such as 53BP1 loss, which rescues the lethal principle of PARP inhibition caused by HRd, is ineffective in rescuing sensitivity in RB1-defective contexts provides an argument against such a view.

We note that while PARPi have been clinically approved for cancers with documented HRd a large amount of preclinical work highlights that HRd is just one mechanism that leads to selective sensitivity to PARPi. The current lack of approved PARPi use in cancers with such other defects is likely explained by the absence of suitable biomarkers for patient identification and/or low prevalence by which the defects arise, posing difficulties to gain regulator-approval.

The fact that there is no correlation of cisplatin sensitivity with RB1 status in the collection of cell lines does not necessarily support a lack of an HR defect. These differences are likely due to the fact that a large number of factors affect cisplatin sensitivity. Therefore, it is not surprising that these cell lines would have greatly different sensitivities to platinum. Additionally, some of the cell lines may have been derived from platinum-resistant tumors.

We agree with the reviewer and we acknowledge this in our writing. However, and overall, the cells of our panel as a whole appear sensitive, as benchmarked using platinum sensitive CAPAN1. The data thus suggest that platinum sensitivity is a general feature of osteosarcoma, or, at least, the osteosarcoma lines that are available for 2D work. The discordance between platinum and PARPi sensitivity is indicative that PARPi sensitivity is not readily predicted by the response to platinum therapy, contrasting the situation in ovarian cancers, and hence, with caution, the data provide some level of preclinical guidance for biomarker informed translation of PARPi in osteosarcoma.

The studies do not show that an RB1-mutation (or deficiency) actually affects the antitumor activity of talazoparib in the xenograft model. Instead, they show that the xenograft grows more slowly – but does not shrink – when treated with talazoparib. This could be true for many tumors, even those that lack known HR defects.

In response to the reviewer’s concern we are now adding *in vivo* xenograft data using RB1-normal CAL72. The results document absence of significant response. The data are shown in **Figure 9b and d**.

Similar findings have been previously reported. A previous publication “Osteosarcoma cells with genetic signatures of BRCAness are susceptible to the PARP inhibitor talazoparib alone or in combination with chemotherapeutics” (PMC5564725) concluded that an HRD-LOH signature occurred in some osteosarcoma cell lines and was associated with sensitivity to talazoparib. These findings, which are somewhat contradictory to findings reported here, were not discussed.

The manuscript noted by the reviewer assesses the hypothesis that HRd, predicted using the frequency of large-scale deletion HRD-LOH as criterium and, independently, loss of genes previously found to identify PARPi sensitivity, predicts PARPi response in osteosarcoma.

The outcome reported is not clear cut. Two cell lines with positive HRD-LOH seem highly sensitive to single-agent talazoparib, while another shows poor sensitivity akin to that of U2OS, that did not score for HRD_LOH. A fourth line, scoring in HRD-LOH as intermediate, also was poorly sensitive. All cell lines featured copy number changes in genes associated with PARPi sensitivity, hence this feature seems not predictive of single-agent sensitivity in the panel. Based on these presented data it would seem therefore that the hypothesis would need to be rejected, namely neither HRD_LOH nor gene mutational profiles predicts single-agent PARPi sensitivity.

In terms of data comparison: Single-agent sensitivity in the manuscript is measured using a short term 72hrs viability assays, and hence the IC50 values reported cannot be directly compared with those from our work, which uses 5 day-drug exposures, a format where sensitivity can be more robustly compared across an extended cell line panels as the effect of cell cycle differences between lines is largely mitigated.

A later figure in the manuscript shows single dose clonogenic survival for two cell lines, including MG63, which we characterise in our study. Albeit the lines in the initial 72-hour assessment were considered sensitive, they appear to lack response to 10 nM talazoparib using clonogenicity. These data agree with our data for MG63, yielding an IC50 of 40 nM, assessed using clonogenicity in our study. Importantly, this IC50 is 20 to 100-fold higher than the IC50 values in RB1-defective lines, and more than 10-fold higher than the IC50 for a *BRCA2*-mutated cell line CAPAN 1. Hence MG63 does not feature as acceptably sensitive according to our analysis.

We felt that overall the scope of the manuscript was peripheral to our work (which does not aim to test for the effect of HRd on sensitivity). However, if the reviewer feels it should be cited we will do so.

Minor comments:

The use of the word “matched” to describe cell lines is potentially misleading. The cell lines that were being compared were from different patients. Depending on your viewpoint, “matched” could mean that the cell lines being compared were quite similar (e.g., a mutation in a specific gene).

We have deleted the word matched. The word relates to the fact that lines with and without RB1 loss were related in histiotype. The context is explained in the prior sentence and hence the word is unnecessary and, we agree, potentially misleading.

In many figures where a representative experiment is shown, it’s not clear how many times the experiment was repeated with a similar result.

We have systematically added this information. All conclusions are based on a minimum of two datasets. A higher number of repeat datasets are available for a number of cell lines, because they were assessed more frequently, with purpose to benchmark consistency of

data over time. We show response curve data for individual dataset, providing exemplary evidence for quality of curve fits. However, summary data in the form of scatterplots for AUC and IC50 are invariably shown for each type of analysis, covering datasets from multiple runs.

We note that all outcomes based on osteosarcoma-derived cell lines are confirmed using cell lines with RB1-targeting modification, and key outcomes are confirmed using orthogonal assay (i.e. day- 5 viability and clonogenic survival). Hence, we are confident that the outcomes presented are genuine and causatively linked to RB1 loss.

Reviewer 3

We thank the reviewer for commenting on our manuscript and their positive appraisal of the work.

The correlation between different RB1 wt or null lines is striking, but a direct role for RB1 was tested in only one cell line (CAL72). Is it possible to add another line for at least some of the assays to add extra confidence that RB1 loss enhances sensitivity? This seems relevant given that depleting RB1 in CAL72 cells increased sensitivity to platinum, which was not reflected in comparison of other RB1 null or wt osteosarcoma cells.

A similar concern was voiced by reviewer 2. We now include data using a second RB1-normal osteosarcoma cell line, 143B, documenting enhanced PARPi sensitivity following RB1 ablation (**shown in Supplementary Figure 4 e-g**) and evidence for PARPi-dependent gamma H2AX accumulation (**shown in Supplementary Figure 7d to h**), both observed using two different shRNAs.

On a related note, is it feasible to ask whether adding RB1 back to RB1 null cells (to levels similar to those in RB1 wt osteo) reduces sensitivity? This would not need to be done in a large number of lines (one or two would suffice), but would add confidence (to the depletion assays above) that RB is the critical determinant. It may be difficult/impossible to create stable lines with RB added back, but might be feasible to use viral vectors to establish a population of transduced cells for the drug study. The authors have done a very nice job correlating RB1 loss to PARP sensitivity, and it would add so much to their paper to confirm the functional connection.

AND

If cell survival with PARPi does, as predicted, fall with RB1 depletion and rise with RB1 restitution, it'd be important to test whether this also reflects difference in DSB induction.

We thank the reviewer for their thoughts. We have attempted such an experiment using conditional (doxycycline-regulated) RB1 expression. We found that RB1- induction consistently leads to a dominant G1 arrest in the RB1-defective osteosarcoma lines, most likely due to the expression of high level p16INK4a observed in all the RB1-defective osteosarcoma lines. It is a recognised phenomenon (the reviewer may be aware) that cancers with loss of RB1 express high p16INK4a, preventing activation of CDK4/6 involved in RB1 inactivation. The cause of the tight correlation might be explained by the fact that growth of cancers with RB1 loss is possible without CDK4/6 activation, and hence without the need to inactivate the INK4 tumour suppressor locus. There are some reports documenting that experimental ablation of p16INK4A in RB1-defective backgrounds is

synthetically lethal, arguing for a cancer evolutionary advantage to retain capability to express p16INK4A.

While the growth arrest we observed in our experiments precluded colony formation and day-5 proliferation assessment, we have collected data for PARPi induced death and gammaH2AX, which both are highly significantly suppressed following RB1 induction. These data are in line with the demonstration that inhibition of cell cycle progression using thymidine prevents death, but they unlikely model the situation of naturally RB1-normal cells, which do not show spontaneous or PARPi-induced G1 cell cycle arrest phenotypes (shown in **Supplementary Figure 7i, k**). We have therefore not included these data in our revised manuscript.

The *in vivo* work is also somewhat limited, again based on a single line, and no comparison of isotype lines +/- RB1. I wouldn't insist on more of these tumor assays given the work required, but the authors should at least acknowledge the serious limitations of their analysis, and note that more rigorous studies are required to test sensitivity of RB1 null vs wt tumors to PARPi *in vivo*.

We are now including experiments using RB1-normal osteosarcoma CAL72, documenting that this *in vitro* resistant line does not significantly respond *in vivo* to PARPi. While it clearly would be logical to extend the work to syngeneic models with engineered RB1 loss, to do such work well would require resources that we would have difficulty to gain access to or justify ethically as aiding patient benefit. Ultimate proof of concept for clinical value of RB1 loss as a biomarker of PARPi sensitivity will require clinical trials in patients. While documentation of efficacy (as we do) generally is considered important to justify such clinical trial to be run, data from engineered preclinical *in vivo* models are not likely to add substantive additional weight.

As suggested by the reviewer we have included the following revised sentence flagging up limitations.

'While detailed studies using isotype-matched cell lines with engineered RB1 loss would be necessary for proof-of-concept of RB1 dependence, the results suffice to document single-agent effects in line with those observed in cell-based experiments, and coherent with such an assumption'.

The Discussion touches on the puzzle as to why PARP inhibitors are so effective in RB1 null osteosarcoma, despite the fact that HR appears to be intact. It could be a bit better developed to summarize different theories as to how PARP inhibitors kill cells, which are nicely summarized by Rose et al (PMID: 33015058). Whether any of these mechanisms is responsible in RB null osteosarcoma is beyond the scope of this paper, but it does seem critical to discuss in more depth. It is surprising that HR-intact cells are so sensitive. In that vein, it was reported recently that another RB1 null cancer, retinoblastoma, also has intact HR, but is hypersensitive to HR inhibition, and a RAD51 inhibitor synergized with agents like topotecan which, like PARP perturb replication progression (Aubry et al, PMID: 32572160). Perhaps all RB1 null cells are hyper-dependent on HR, and that dependency could contribute to PARP sensitivity? The authors could explore the possibility that PARP.

We have expanded the discussion as suggested by the reviewer and cite both these publications and further primary relevant work. It has been helpful to reflect on this matter, in particular highlighting that albeit PARPi are confined to use in patients with HRd synthetic PARPi hypersensitivity is not confined to this mechanistic lesion but extends to a substantive number of other defects, unrelated to HRd. It is to be suspected that clinical translation in these contexts has not arisen because detection of defects in the respective processes is not readily feasible in clinical samples or defects arise infrequently or in disease without unmet

clinical need, making it difficult to obtain clinical proof of concept. This contrasts with RB1 loss in cancers, which should be readily identifiable using routine clinical materials, and arises with sufficient frequency and in disease with pressing medical need, including osteosarcoma,

Minor

Typos

Fig 1 a/c/e legend - “back” should be “black”.

Fig 8e graph “Vehcile” = Vehicle

These errors have been corrected

Reviewers' Comments:

Reviewer #1:

Remarks to the Author:

The authors provided additional data requested for another cell line, for which I am grateful.

Because these results are essentially a correlation noted in a relatively small number of samples, secondarily strengthened by forward-genetic experiments in (now) 2 cell lines, but no reverse genetic experiments (the restoration of RB1, that--not surprisingly--is not technically very feasible), the mechanistic insight is very thin.

This does not claim to be a mechanistic paper, however. It claims to show translational impact as identifying a biomarker for PARPi sensitivity among some osteosarcomas. This is an interesting and potentially important idea. It seems that it now requires a deeper dive clinically in a larger number of PDXs or some other way of testing their efficacy in a larger population of RB1-wt and RB1-mut osteosarcomas, to see how well the pattern follows in larger populations.

One very minor fix that is necessary:

RECQL4, BLM, and WRN are not frequently mutated, as identified in osteosarcoma genomics. That notation should be removed from the introduction. Mutations in these RECQL4 and these other helicases drive genetic syndromes, some of which include an increased risk for the development of osteosarcoma. Osteosarcomas arising in non-syndromic populations have not been found to have frequent mutations in these genes.

Reviewer #2:

Remarks to the Author:

The authors have sufficiently addressed my concerns and comments, and I believe that the study is important and timely. However, a few minor points should be addressed.

1. The section entitled "Mechanism of PARPi sensitivity in RB1-defective osteosarcoma" should be re-titled. The studies shown do not identify a mechanism that explains why RB1-deficient cells are more sensitive to PARP inhibitors.

2. The sentence on page 9 that states: "53BP1 loss broadly restores HR following BRCA1 loss and further loss of Ataxia telangiectasia mutated (ATM) kinase 32, required during HR for events upstream as well as downstream of RAD51 recruitment cite 33 34" does not make sense and has grammatical errors. In addition, it is not clear why ATM is being discussed here. Finally, telangiectasia is misspelled.

3. The sentence on page 10 that states: "Together these data argue that defects in are not associated with canonical features of HRd/ BRCAness and hence that PARPi sensitivity in RB1-defective osteosarcoma is mechanistically distinct from and not explained by outright inability to engage HRbased DNA repair" has an error. Perhaps it should say "Together these data argue that defects in HR are not associated with...."

4. There are multiple typos, syntax, and grammar errors.

5. Finally, the Discussion lacks depth, and the points are not clearly articulated.

Reviewer #3:

Remarks to the Author:

The authors have addressed all my major concerns. I thank them for their hard work in addressing the comments.

One minor textual issue that should be corrected: The Discussion added mention of the work by

Aubry et al on the sensitivity of retinoblastoma to HR disruption. It states: "Lethality was seen in retinoblastoma-derived cell lines with RB1 defect but not an RB1 normal MYCN amplified line, suggesting a causal role of RB1 loss in the selective lethality" (page 16). My understanding is that the sensitivity was common to RB1^{-/-} and MYCN amplified tumors. Aubry et al stated that their screen exposed: "essential roles for DNA-repair proteins in both tumor subtypes". The author's subsequent comment that: "frank HRd may not be acceptable in RB1 mutated backgrounds" is, however, still valid and worth stating.

Point by point reply to Reviewers

Reviewer #1

One very minor fix that is necessary:
RECQL4, BLM, and WRN are not frequently mutated

Thank you for pointing this out. We agree that the statement is incorrect as made, and have revised, now reading:

Emerging osteosarcoma genomics data reveal the prominent presence of deleterious mutations in the known tumour suppressors including TP53, RB1, ATRX and CDKN2A.

Reviewer #2

- 1. The section entitled “Mechanism of PARPi sensitivity in RB1-defective osteosarcoma” should be re-titled.**

We have revised this headline now stating:

“Determinants of PARPi sensitivity in RB1-defective osteosarcoma”

- 2. The sentence on page 9 that states: “53BP1 loss broadly restores.. does not make sense and has grammatical errors. In addition, it is not clear why ATM is being discussed here. Finally, telangiectasia is misspelled**

Thank you. We simplified this sentence, now reading,

We further assessed the impact of the 53BP1 loss (Supplementary Figure 5b-o). 53BP1 ablation broadly abolishes PARPi sensitivity consequent to BRCAness/HRd {Hong, 2016 #95} {Jaspers, 2013 #114} {Misenko, 2018 #115} {D’Andrea, 2018 #116}. While shRNA....

- 3. The sentence on page 10 that states: “Together these data argue that defects are not associated with canonical features of HRd/ BRCAness.**

‘in RB1” was omitted in this statement.

The statement has been corrected to read “....defects in RB1 are not associated....”

- 4. multiple typos, syntax, and grammar errors**

text checked by independent reader; errors corrected where found.

- 5. the Discussion lacks depth, and the points are not clearly articulated.**

We have revisited the discussion and undertaken revision to increase clarity. Our aim in the discussion had been to position our work with respect to existing mechanistic knowledge and clinical activity, and to highlight tenable conclusions and the significance of the work.

Reviewer #3

6. **minor textual issue that should be corrected**, sensitivity is common to RB1-/- and MYCN amplified tumors

Thank you for identifying the misunderstanding on our site. We have amended the statement now reading

Intriguingly, a recent functional genomics screen identified HRd as selectively lethal in cell lines derived from retinoblastoma, a tumour primarily initiated by RB1 loss. While it is not yet clear whether a similar selective lethality exists in other cancer types with RB1 loss, the results as they stand could suggest that frank HRd is not acceptable in RB1-mutated backgrounds {Aubry, 2020 #103}.